# Stabilizing a metalloid {$Zn_{12}$} unit within a polymetallide environment in [$K_2Zn_{20}Bi_{16}$]$^{6-}$

Armin R. Eulenstein [1,2,6], Yannick J. Franzke [3,5,6], Patrick Bügel[4], Werner Massa [1], Florian Weigend [1,4✉] & Stefanie Dehnen [1,2✉]

The access to molecules comprising direct Zn–Zn bonds has become very topical in recent years for various reasons. Low-valent organozinc compounds show remarkable reactivities, and larger Zn–Zn-bonded gas-phase species exhibit a very unusual coexistence of insulating and metallic properties. However, as Zn atoms do not show a high tendency to form clusters in condensed phases, synthetic approaches for generating purely inorganic metalloid $Zn_x$ units under ambient conditions have been lacking so far. Here we show that the reaction of a highly reductive solid with the nominal composition $K_5Ga_2Bi_4$ with $ZnPh_2$ at room temperature yields the heterometallic cluster anion [$K_2Zn_{20}Bi_{16}$]$^{6-}$. A 24-atom polymetallide ring embeds a metalloid {$Zn_{12}$} unit. Density functional theory calculations reveal multicenter bonding, an essentially zero-valent situation in the cluster center, and weak aromaticity. The heterometallic character, the notable electron-delocalization, and the uncommon nano-architecture points at a high potential for nano-heterocatalysis.

[1] Fachbereich Chemie, Philipps-Universität Marburg, Hans-Meerwein-Str. 4, 35032 Marburg, Germany. [2] Wissenschaftliches Zentrum für Materialwissenschaften (WZMW), Philipps-Universität Marburg, Hans-Meerwein-Str. 6, 35032 Marburg, Germany. [3] Institute of Physical Chemistry, Karlsruhe Institute of Technology (KIT), Kaiserstr. 12, 76131 Karlsruhe, Germany. [4] Institute of Nanotechnology, Karlsruhe Institute of Technology, Hermann-von-Helmholtz-Platz 1, 76344 Eggenstein-Leopoldshafen, Germany. [5]Present address: Fachbereich Chemie, Philipps-Universität Marburg, Hans-Meerwein-Str. 4, 35032 Marburg, Germany. [6]These authors contributed equally: Armin R. Eulenstein, Yannick J. Franzke. ✉email: florian.weigend@chemie.uni-marburg.de; dehnen@chemie.uni-marburg.de

The use of nontoxic elements for applications in chemical synthesis, as well as for innovative and harmless materials is highly desirable. Zinc and bismuth are such nontoxic metals, hence enjoying high reputation in this context. However, especially zinc is predominantly found in the +II oxidation state in its compounds, owing to the inherently high stability of the corresponding $3d^{10}$ electronic configuration. For this, one branch of contemporary zinc chemistry is dedicated to the formation of low-valent zinc compounds for enhancing its chemical reactivity. A straightforward strategy toward low-valent metal compounds includes the formation of metal–metal bonds, hence approaching the metallic state eventually, yet in molecular compounds. Most metals in the periodic table of the elements have been known to form metal–metal-bonded molecules, with some exceptions that represent a particular challenge to synthetic chemists. The first species with a Mg–Mg bond, for instance, was reported in 2007 (ref. [1]), and also the synthesis of the first compound comprising a covalent Zn–Zn bond was reported as late as in 2004 for Cp*Zn–ZnCp* (ref. [2]). In the meantime, further species of the type RZn–ZnR were added, including a compound with $Ge_9$ cages replacing the organic substituents[3], as well as its triangular Zn(I)/Zn(II) derivative $[Cp_3Zn_3]^+$ (ref. [4]), and the lower-valent variant Cp*Zn–Zn–ZnCp*, including a formal Zn(0) atom[5]. In-depth investigation of such species during the past 15 years indicated extraordinary reactivity and activation properties, which opened up another branch of reactive metalloid compounds with relatively benign metals[3,6–10]. However, metals that do not tend to readily form metal–metal bonds are also unlikely to form larger aggregates, so-called metalloid clusters, under ambient temperatures and pressures—which immediately raises the exciting question, how such virtually inaccessible clusters would behave, if an experimental access was actually found.

Gas-phase studies of zinc clusters that were generated in a magnetron sputter gas aggregation source indicated unusual electronic properties, which reveals the coexistence of insulating and metallic properties[11,12]. This has been a tremendous inspiration for synthetic groups that yearned for the isolation of $Zn_x$ aggregates with $x > 3$ in condensed phases. First, discrete clusters containing organozinc units with up to ten Zn atoms, were reported very recently by the Fischer group, in which the $Zn_x$ subunits are protected and kinetically stabilized by organic substituents like alkyl groups or Cp* (refs. [13,14]). Besides this, Zn clusters were only found as guests in Zeolite X (ref. [15]) or other porous materials[16], and they were identified as subunits in neat intermetallic phases[17,18]. Hence, clusters with ligand-free Zn atoms, which would enable direct access to the Zn sites in reactivity studies, are still lacking.

Heterometallic and intermetalloid clusters can be viewed as molecular mimics of heterometallic materials[19–22]. Usually, the number of heteroatomic bonds is maximized in such clusters for a gain of bond energy through heteropolar interactions, but there are a few exceptions to this, such as $[Sb_3Au_3Sb_3]^{3-}$ (ref. [23]), or $(Ge_4Bi_{14})^{4-}$ (ref. [24]), that exhibit a clear segregation of the involved atom types. Notably, both clusters refer to elemental combinations that are virtually immiscible in the solid state. Hence, we aimed at the formation of larger metalloid Zn aggregates by synthesizing heterometallic clusters with elemental combinations lacking size match.

Only three heterometallic clusters with Zn and Bi atoms have been reported to date: the binary intermetalloid cluster anion $[Zn@Zn_8Bi_4@Bi_7]^{3-}$ (ref. [25]), and the related ternary species $[Zn@Zn_5Tt_3Bi_3@Bi_5]^{4-}$ (Tt = Sn (ref. [26]), Pb (ref. [27])). In the first case, the Zn atoms are found in a triangle and two dumbbells in the cluster shell. In the second, they show some segregation, with five Zn atoms forming weak interaction within a five-membered ring. The quoted clusters were obtained by reactions of $K_5Bi_4$ or

$[K(crypt-222)]_2(Tt_2Bi_2)\cdot en$, respectively, with $ZnPh_2$ (crypt-222 = 4,7,13,16,21,24-hexaoxa-1,10-diazabicyclo[8.8.8]hexacosan; Tt = Sn, Pb; en = ethane-1,2-diamine). In both cases, population analyses show that the natural charges of the involved atoms are mostly +1 (Zn) and −1 (Bi). We assumed that further aggregation of Zn atoms might be obtained under conditions that let the reduction of $ZnPh_2$ proceed further toward the finally metallic Zn. Recent investigations using the binary anion $(GaBi_3)^{2-}$ indicated a tendency of this anion to release elemental $Ga^0$ during heterometallic cluster formation, hence releasing two electrons per formula unit in situ that can be used to form metalloid clusters[28,29]. This precursor compound, however, did not prove suitable for reactions with $ZnPh_2$, as no change of the reactants was observed. A ternary mixture of the nominal composition $K_5Ga_2Bi_4$ (ref. [30]) in contrast (with five negative charges per six metal atoms instead of two negative charges per four atoms), seemed to be reductive enough: indeed, the precursor is able to reduce $ZnPh_2$ to form the heterometallic cluster $[K_2Zn_{20}Bi_{16}]^{6-}$, with an essentially zero-valent situation at the Zn atoms in the cluster center and distinct electron delocalization within its extraordinary molecular architecture.

Here, we report an approach to molecular Zn clusters via heterometallic cluster synthesis that was developed by systematic investigations of potential elemental combinations in corresponding reactions in solution.

## Results

**Synthesis and characterization of $[K(crypt-222)]_6[K_2Zn_{20}Bi_{16}]$ (1).** The reaction of a solid with the nominal composition $K_5Ga_2Bi_4$ with $ZnPh_2$ in a 1:2.5 molar ratio in en/crypt-222 at room temperature, subsequent filtration and layering of the solution with toluene affords thin, black crystals of the most probable composition $[K(crypt-222)]_6[K_2Zn_{20}Bi_{16}]$ (**1**; Supplementary Fig. 1). An X-ray structure analysis reveals the presence of the cluster anion $[K_2Zn_{20}Bi_{16}]^{6-}$ (**1a**), comprising a homoatomic subunit of 12 directly bonded Zn atoms. Despite being obtained from solution, **1a** represents a ligand-free nanocluster comprising 36 metal atoms, hence exceeding the number of metal atoms reported recently for $[Rh_3@Sn_{24}]^{5-}$ (ref. [31]).

The cluster anion **1a** is shown in Fig. 1. It possesses an idealized point group symmetry of $D_{2d}$, which is reduced to $C_{2v}$ in the crystal structure due to slight distortions. The $C_2$ axis runs through two $K^+$ cations that are coordinated above and beneath a bimetallic, crown-like $[Zn_{20}Bi_{16}]^{8-}$ cluster; besides K1 and K2, the two vertical mirror planes include Bi2, Bi4, Zn5 and Bi5, Bi6, Zn6, respectively. The outer dimensions of the nanocluster **1a** are 12.60 Å (Bi2···Bi2$^c$), 12.55 Å (Bi5···Bi5$^c$), 11.36 Å (Bi1···Bi1$^c$), and 6.86 Å (K1···K2). During the reaction, all Zn atoms released their Ph groups to be ligand-free atoms within the cluster. Unusually high, yet reasonable, displacement parameters in the structure of **1a** point to orientational disorder of the metal atoms over close positions. Obviously, this also involves disorder of the cations, which in the presence of the heavy atoms of the anions hampers the localization of the atomic positions of the light atoms of the cryptate ligands from the Fourier map. Yet, the spatial demand around the K atoms (Supplementary Fig. 2 and Supplementary Discussion) and the features of the electron density distribution in the Fourier maps agrees with the assumption of $[K(crypt-222)]^+$ cations. To reduce the impairment of the refinement of the anionic cluster by an incomplete model, the influence of these parts was detracted from the data by the back Fourier transform method[32]. The crystal data and experimental parameters of the structure determination of **1** (CCDC 1969162) are collected in Supplementary Table 1. Figure 1c, d illustrates the packing of cations and anionic clusters. The latter are arranged in two types

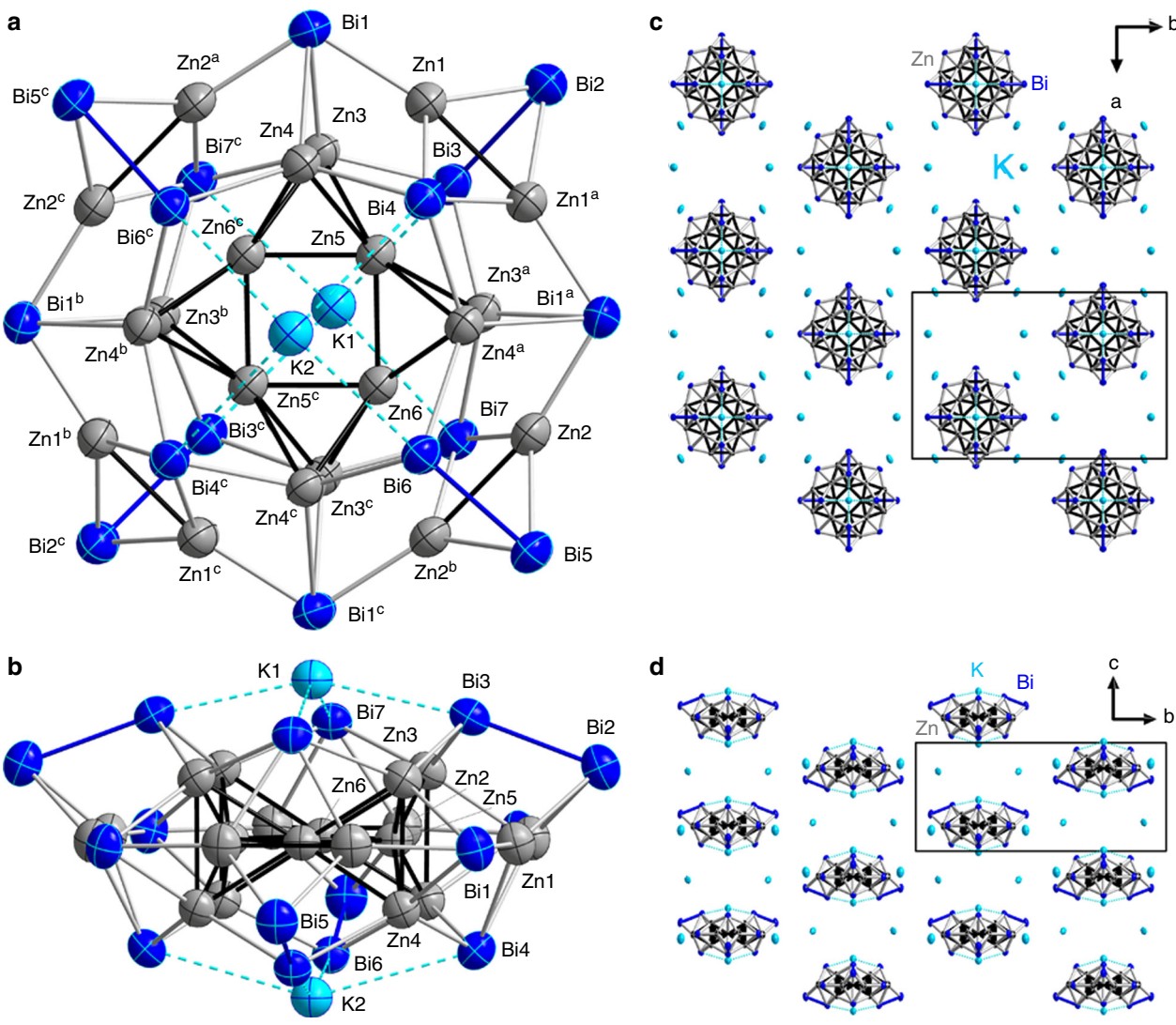

**Fig. 1 Crystal structure of [K(crypt-222)]₆[K₂Zn₂₀Bi₁₆] (1). a** View onto the oblate molecule [K₂Zn₂₀Bi₁₆]⁶⁻ (**1a**) possessing idealized $D_{2d}$ symmetry, which is reduced to crystallographic $C_{2v}$ symmetry. **b** Side view, upon an additional rotation about the $C_2$ axis running through K1 and K2 (by ~45° with regard to the orientation above). Displacement ellipsoids are drawn at 30% probability. Selected distances [Å]: Zn1–Zn1ᵃ 2.756(5), Zn2–Zn2ᵇ 2.831(5), Zn3–Zn4 2.812(4), Zn5–Zn(3,4) 2.664(3), 2.682(3), Zn6–Zn(3ᵃ,4ᵃ) 2.681(3), 2.674(3), Zn5–Zn6 2.544(3); Zn1···Zn(3,4) 2.879(4), 2.904(4); Bi1···Zn (3,4) 2.893(3), 2.885(2); Bi1–Zn(1,2ᵃ) 2.683(3), 2.692(3), Bi2–Zn1 2.701(3), Bi5–Zn2 2.699(3), Bi3–Zn3 2.818(2), Bi4–Zn4 2.798(2), Bi7–Zn3ᵃ 2.809(2), Bi6–Zn4ᵃ 2.816(2), Bi4–Zn1 2.930(3), Bi7–Zn2 2.921(3); Bi2–Bi3 3.053(2), Bi5–Bi6 3.0414(16); K1···Bi3,3ᶜ 3.528(2), K2···Bi4,4ᶜ 3.741(3), K2···Bi6,6ᶜ 3.533 (2), K1···Bi7,7ᶜ 3.725(3). Symmetry codes: ᵃ ½ − x, y, z; ᵇ x, ½ − y, z; ᶜ ½ − x, ½ − y, z. **c** View of the packing of [K₂Zn₂₀Bi₁₆]⁶⁻ anions and K⁺ cations along the crystallographic c axis. **d** View of the packing of anions and cations along the crystallographic b axis. C, N, and H atoms are not shown.

of pillars along the crystallographic c axis, with inverse orientations of the clusters with respect to the (idealized) $S_4$ axis, and with the clusters shifted by c/2 relative to each other.

The most intriguing feature within the [Zn₂₀Bi₁₆]⁸⁻ cluster structure is the assembly of 12 Zn atoms (Zn3–Zn6 and symmetry equivalents) in the cluster center. These atoms are arranged in four corner-sharing tetrahedra that form a nearly undistorted inner Zn₄ square (angle Zn6–Zn5–Zn6ᶜ 90.48(14)°). The tetrahedra are not regular, with a short inner Zn5–Zn6 edge (2.544(3) Å), somewhat longer contacts to the outer edges (2.664 (3)–2.681(3) Å) and an elongated outer Zn3–Zn4 edge (2.812(4) Å). However, the Zn–Zn–Zn angles differ only slightly from ideal values (56.72(9)–63.48(8)°). As Ga and Zn atoms cannot be easily distinguished by means of common X-ray diffraction, the question whether Ga atoms might be involved in the structure instead of or in addition to Zn atoms was clarified by energy-dispersive X-ray spectroscopy (EDS) on single crystals of **1**

(Supplementary Figs. 3, 4, and Supplementary Table 2), and by DFT calculations (vide infra). Both clearly rule out the presence of any Ga atoms in the cluster anion **1a**.

The inner {Zn₁₂} unit is embedded in a macrocycle consisting of the eight remaining Zn atoms (Zn1, Zn2, and symmetry equivalents) and 16 Bi atoms (Bi1–Bi7 and symmetry equivalents), to which it is bonded by Zn–Bi contacts (2.798(2)–2.893 (3) Å). The atoms of the {Zn₈Bi₁₆} moiety are connected by different metal–metal bonds: Zn–Zn (2.756(5), 2.831(5) Å), Bi –Bi (3.053(2), 3.0414(16) Å) and Zn–Bi (2.683(3)–2.930(3) Å). The nature of the interatomic interactions were studied in detail by means of quantum chemical calculations.

**Quantum chemical investigation of the bonding in [K₂Zn₂₀Bi₁₆]⁶⁻ (1a).** We optimized[33,34] the geometric structure at DFT level (TPSS/dhf-TZVP/grid m3). The calculated molecular

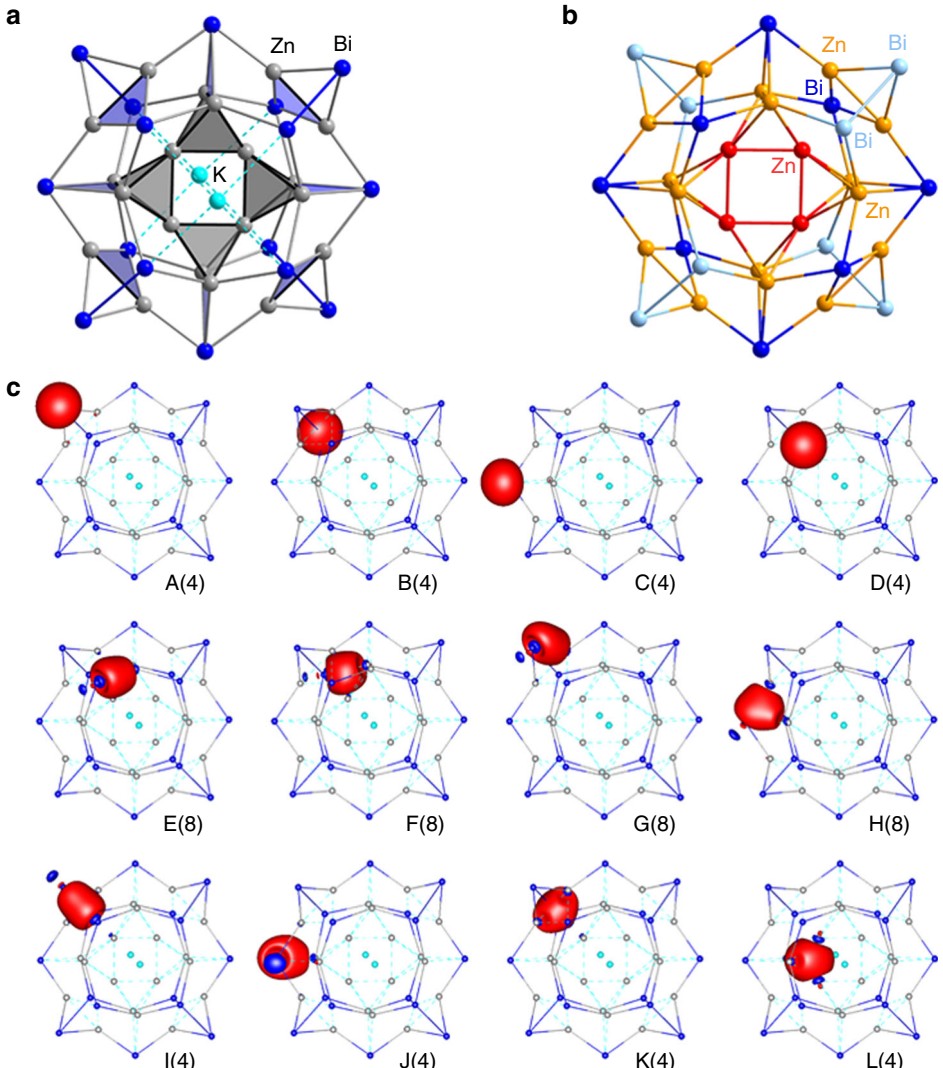

**Fig. 2 Chemical bonding in [K$_2$Zn$_{20}$Bi$_{16}$]$^{6-}$ (1a). a** Illustration of the bonding situation from a Boys localization procedure for the valence orbitals followed by calculation of atomic Mulliken contributions to each localized molecular orbital (LMO) and neglecting contributions <20%. Resulting two-center bonds are displayed by solid lines, three-center and four-center bonds by planes and polyhedrons, respectively. **b** Results of a NPA performed on [Zn$_{20}$Bi$_{16}$]$^{8-}$, with atoms that possess charges of the given ranges drawn in the following colors: +0.0...+0.1 (red), +0.7...+0.9 (yellow), −1.0...−1.1 (light blue), and −1.5...−1.6 (dark blue). **c** LMOs A–D representing the lone pairs at each of the Bi atoms, LMOs E–J representing Zn–Bi and Bi–Bi two-center bonds (with H and J showing significant contributions from one further atom), and LMOs K and L representing closed Zn–Zn–Bi three-center bonds and four-center bonds within the four Zn$_4$ subunits forming the central {Zn$_{12}$} unit. Amplitudes are drawn at 0.05 atomic units (a.u.), numbers refer to the number of equivalent LMOs of the shown type.

orbitals show a significant HOMO–LUMO gap of 1.4 eV, but no further significant gaps in the region of frontier orbitals, which indicates that the total electron number and thus the assumed composition are correct (an isoelectronic alternative, including Ga atoms in the cluster center would require a lower overall charge, hence "[K$_2$Zn$_{16}$Ga$_4$Bi$_{16}$]$^{2-}$", which is clearly ruled out by the detection of six additional K$^+$ counterions in the crystal structure of **1**, and the corresponding electron density representing six disordered cryptand molecules in the voids between the anions). For a plausible assignment of bonds in [K$_2$Zn$_{20}$Bi$_{16}$]$^{6-}$, we carried out a Boys localization procedure[35] for the 164 valence orbitals and calculated the atomic Mulliken contributions[36,37] to each localized molecular orbital (LMO). The results are illustrated in Fig. 2. For each of the LMOs, one representative is shown in Fig. 2c. When neglecting atomic contributions <20%, this reveals the following picture. A total of 116 LMOs represent one-center contributions: 20 × 5 LMOs for the Zn(d) orbitals and 16 LMOs

A–D, representing one lone pair for each of the 16 Bi atoms. A total of 32 LMOs E–H represent Zn–Bi two-electron-two-center bonds (straight lines): eight Zn–Bi bonds within the upper inner ring (E), another eight within the lower inner ring (F), and 16 within the outer ring (G, H). All of these bonds are polarized, which is evident from the Mulliken contributions to the LMOs, which amount to 52–66% for Bi and to 22–41% for Zn (for a more refined picture we note that the two-center bonds in the outer ring show contributions from the Zn atoms in the inner rings (typically 10%) and vice versa, in particular LMO H). Among the remaining 16 LMOs, 12 connect the outer ring with the two inner rings: 4 Bi–Bi bonds (I), 4 three-center bonds (blue triangles, K), and four more bonds (blue triangles, J), which also may best be viewed as three-center bonds (the shown representative as well as two of its equivalents binds mainly to the lower inner ring, whereas the fourth binds mainly to the upper inner ring. This is an unphysical break of symmetry from the localization procedure, and a less

strict interpretation as three-center bonds appears more reasonable). The inner four Zn atoms are involved exclusively in the finally remaining four LMOs (gray tetrahedra, L) representing four-center bonds that connect the unique $Zn_{12}$ unit within **1a**.

A natural population analysis (NPA)[38] performed on the optimized structure of **1a** (Fig. 2b) yields charges of +0.05 for the inner four Zn atoms and +0.71 for the other eight Zn atoms. The latter is a typical value for an oxidation state of +I (compare, for instance, ZnCl: +0.71, ZnBr: +0.67), so this unit as a whole is clearly low valent. We note in passing that this {$Zn_{12}$} unit as an isolated species is stable even as a neutral species. Structure optimizations carried out for charges $q = 0$, +2, +6 (Supplementary Fig. 5) show that the structure of the {$Zn_{12}$} moiety persists in an isolated form, for $q = 0$ even with a reasonable HOMO–LUMO gap of 1.6 eV, indicating the general stability of this cluster unit. However, a second local minimum was found that is less stable for {$Zn_{12}$}$^{±0}$ (+57 kJ/mol) and {$Zn_{12}$}$^{2+}$ (+9 kJ/mol), but is favorable for {$Zn_{12}$}$^{6+}$ (−138 kJ/mol). In summary, the calculations confirm **1a** as being a cluster with a large low-valent subunit of group ten metals known to date. Efforts to experimentally probe the atomic charges by means of X-ray photoelectron spectroscopy (XPS) failed so far owing to the high sensitivity of the very thin crystals, which spontaneously oxidized during sample preparation and thereupon produced XPS signals of $Zn^{2+}$ and $Bi^{3+}$ only. Further studies are underway to first of all increase the crystal size.

**Relationship of the {$Zn_8Bi_{16}$}$^{q−}$ unit in 1a to porphine.** Remarkably, the {$Zn_8Bi_{16}$}$^{q−}$ unit ($q = 8...14$ embedding {$Zn_{12}$}$^{±0...6+}$) exhibits a certain relationship with the organic macrocycle porphine, $C_{20}N_4H_{14}$: both rings possess 24 ring atoms, and a similar number of $s$ and $p$ valence electrons (104 for {$Zn_8Bi_{16}$}$^{8−}$ or 110 for {$Zn_8Bi_{16}$}$^{14−}$, vs. 114 for $C_{20}N_4H_{14}$). In both cases, five-atomic units ($Zn_2Bi_3$ vs. $C_4N$) are connected by a one-atom bridge (Bi vs. C), with lone pairs in **1a** replacing the H substituents of porphine. In contrast to porphine, the {$Zn_8Bi_{16}$}$^{q−}$ metallacycle is not known as a separate entity, but both macrocycles are capable of accommodating metal atoms or ions. Of course, owing to the very different nature of the involved elements, the detailed behavior of both macrocycles differs. Very obviously, owing to the larger atomic sizes, the {$Zn_8Bi_{16}$}$^{q−}$ tire-shaped unit can accommodate 12 Zn atoms, while porphine and its porphyrin derivatives usually coordinate one single ion only. Furthermore, with Bi2–Bi7 (and symmetry equivalents) being located above or beneath the plane defined by the other twelve atoms (Bi1, Zn1, Zn2, and symmetry equivalents), the five-atom unit in **1a** is not planar. It represents a nearly planar $Zn_2Bi_2$ diamond that is inclined with respect to the molecule's equatorial plane, with Bi3 and Bi6 (and symmetry equivalents) being further exposed by binding exclusively to Bi2 and Bi5, respectively (and symmetry equivalents). The reason for this exceptional architecture of the {$Zn_8Bi_{16}$}$^{q−}$ moiety is found (a) in the covalently bonded Zn1–Zn1′ and Zn2–Zn2′ pairs, and (b) in the way the macrocycle embeds the {$Zn_{12}$} unit. The latter might be understood as the natural way of how a very flexible, porphine-like macrocycle with the respective elemental combination and electron count can structurally respond to this uncommon guest moiety. The high flexibility of purely inorganic mimics of porphine was recently shown on the example of [$Hg_8Te_{16}$]$^{8−}$ (ref. [39]). However, while the latter possesses 120 valence electrons, hence more valence electrons than porphine, the electron count in {$Zn_8Bi_{16}$}$^{q−}$ is lower than that in the organic macrocycle. Thus, the porphine-related arrangement of atoms observed in **1a** can be viewed as an electron-poor variant of the quoted 24-atom macrocycles. To examine the applicability of this thought experiment,

we were interested to see whether the polymetallide unit would also show all-metal aromaticity, such as observed for several smaller polymetallic ring systems[40–42]. To study aromaticity based on the magnetic criterion, we calculated the magnetically induced current density of the [$Zn_{20}Bi_{16}$]$^{8−}$ cluster based on the magnetic criterion[43,44]. This was done with GIMIC[45,46], using the response to the magnetic field obtained from TURBOMOLE (for details, see the Supplementary Discussion)[47]. The cluster sustains a net diatropic ring current of 0.43–7.0 nA/T, which is obtained upon integration along a plane perpendicular to the molecular plane and parallel to the external magnetic field (Supplementary Figs. 6 and 7 and corresponding discussion). Remarkably, this is about a fifth of the ring current calculated for porphine (25.4 nA/T at the same computational level) or zinc porphyrine (25.1 nA/T), and about a third of the value calculated for benzene (11–15 nA/T)[43,44]. We note that the strength of the ring current depends on the number of electrons participating, the surface, and the topology. It is not a direct measure of aromaticity[44]. Furthermore, the nucleus-independent chemical shift (NICS) approach[48] was utilized. The NICS values of [$K_2Zn_{20}Bi_{16}$]$^{6−}$ and [$Zn_{20}Bi_{16}$]$^{8−}$ are −4.2 and −4.4 ppm, respectively. For comparison, we obtain −8.0 ppm for benzene and −14.6 ppm for porphine. Considering the core electrons in a scalar-relativistic all-electron theory results in −3.6 ppm for [$K_2Zn_{20}Bi_{16}$]$^{6−}$ and −4.2 ppm for [$Zn_{20}Bi_{16}$]$^{8−}$. Thus, the weak aromaticity of the [$K_2Zn_{20}Bi_{16}$]$^{6−}$ and [$Zn_{20}Bi_{16}$]$^{8−}$ cluster in **1a** is notable—in contrast to the properties of the (topologically more porphine-like) [$Hg_8Te_{16}$]$^{8−}$, which exhibits essentially no global ring current (0.24 nA/T)[39], and an insignificant NICS value of 1.3 and 1.2 ppm, respectively.

**Coordination properties of the {$Zn_{20}Bi_{16}$}$^{8−}$ polyanion in 1a.** Another point worth mentioning is the fact that the oblate {$Zn_{20}Bi_{16}$}$^{8−}$ polyanion in the trimetallic cluster **1a** coordinates two K$^+$ ions, in an inverse-sandwich-type manner, at Bi⋯K distances of 3.528(2) or 3.533(2) Å (Bi(3,6)⋯K), and 3.741(3) or 3.725(3) Å (Bi(4,7)⋯K). Figure 3 illustrates the calculated electrostatic potential with and without coordination of K$^+$. The polyanion {$Zn_{16}Bi_{20}$}$^{8−}$ bears a relatively even (negative) electrostatic potential, and the four Bi atoms that are exposed in the inner ring of the cluster (Bi3, Bi4, Bi6, and Bi7) are attractive enough to trap the K$^+$ cations, which polarize these Bi atoms upon coordination.

Notably, the K$^+$ ions prefer this site although an excess of the cation-sequestering agent crypt-222 was present during the formation and crystallization of the title compound. This is a rare observation, which was reported for heterometallic clusters in a few cases only, [$NbAs_8$]$^{3−}$ (ref. [49]), [($MesCu)_2Ge_4$]$^{4−}$ (ref. [50]), [$M_{1−x}@Sn_9$]$^{4−}$, (M/$x$ = Ni/0 (ref. [51]), Co/ ≈ 0.32 (ref. [52])), and [$Au_3Ge_{45}$]$^{9−}$ (ref. [53]). Calculated energies of exchange reactions of **1a** with crypt-222 or 18-crown-6 (18c6), according to Eqs. (1)/(2) and (3)/(4), clearly indicate that the cluster would lose its K$^+$ cations to the cation-sequestering agents, if considered as an isolated species (in kJ/mol): −182 [Eq. (1)], −132 [Eq. (2)], −126 [Eq. (3)], and −76 [Eq. (4)].

$$[K_2Zn_{20}Bi_{16}]^{6−} + \text{crypt-222} \rightarrow [KZn_{20}Bi_{16}]^{7−} + [K(\text{crypt-222})]^+, \quad (1)$$

$$[KZn_{20}Bi_{16}]^{7−} + \text{crypt-222} \rightarrow [Zn_{20}Bi_{16}]^{8−} + [K(\text{crypt-222})]^+, \quad (2)$$

$$[K_2Zn_{20}Bi_{16}]^{6−} + 18c6 \rightarrow [KZn_{20}Bi_{16}]^{7−} + [K(18c6)]^+, \quad (3)$$

$$[KZn_{20}Bi_{16}]^{7−} + 18c6 \rightarrow [Zn_{20}Bi_{16}]^{8−} + [K(18c6)]^+. \quad (4)$$

The same holds for many other cations that were tested this way, also with Sb or As atoms replacing Bi atoms in **1a** (Supplementary Tables 3–8 and Supplementary Equations (1)–(7). However, in the crystalline state, the two K$^+$ cations remain

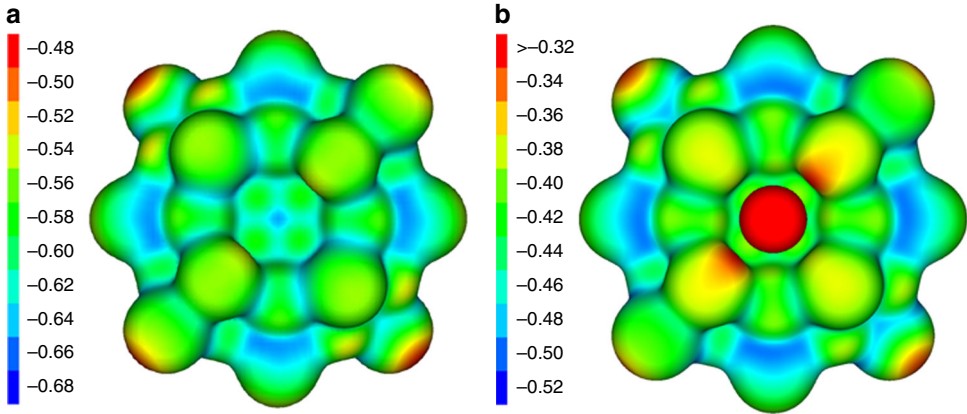

**Fig. 3 Electrostatic potential of calculated cluster anions. a**, Electrostatic potential of $[Zn_{20}Bi_{16}]^{8-}$. **b**, Electrostatic potential of $[K_2Zn_{20}Bi_{16}]^{6-}$ (**1a**). Values are given in atomic units (a.u.).

coordinated—most probably to reduce the overall charge and allow for the formation of single crystals along with six (instead of eight) [K(crypt-222)]$^+$ counterions.

## Discussion

We report on the targeted synthesis of a salt of the heterometallic cluster anion $[K_2Zn_{20}Bi_{16}]^{6-}$ (**1a**), comprising a homoatomic subunit of 12 Zn atoms, and at the same time a large molecular architecture, involving 20 Zn and 16 Bi atoms. As shown by quantum chemical investigations, an inner {$Zn_{12}$} unit in **1a** represents a really metalloid zinc cluster, which is held together by four-center bonding exclusively. This unit is embedded in a polymetallide {$Zn_8Bi_{16}$}$^{q-}$ ring ($q = 8...14$), to which it is connected by three-center and two-center bonds. In striking contrast to the large number of metalloid clusters of other electron-rich $d$-block metals, like the coinage metals Ag and Au, only a few low-valent organozinc clusters were reported to date. **1a** does not only add to this rare class of compounds, it is at the same time an example of a purely inorganic metalloid group 12 cluster within an isolable compound in condensed phase. No low-valent clustering has been reported to date for Cd or Hg, which again sets the cluster **1a** apart from all known compounds involving such elements. Besides these uncommon features, the anion shows a relationship with the organic, aromatic macrocycle porphyine: the 24-membered {$Zn_8Bi_{16}$}$^{q-}$ unit that embeds the inner {$Zn_{12}$} unit possesses a similar valence electron count, and also topological similarities. Notably, the cluster also shows weak aromaticity, indicated by the occurrence of a global ring current that was calculated to be about a fifth of the value calculated for porphine. Weak aromaticity is also suggested by the NICS approach. Furthermore, the disc-shaped anion coordinates two K$^+$ cations in an inverse-sandwich-type manner to reduce the overall charge of the cluster anion, and finally crystallizes as [K(crypt-222)]$_6$[K$_2$Zn$_{20}$Bi$_{16}$] (**1**). We envisage using this monodisperse metal nanocluster, with its heterometallic architecture and its uncommon electronic features in reactivity studies, and eventually nano-heterocatalysis.

## Methods

**General synthesis details**. All manipulations and reactions were performed under dry Ar atmosphere using standard Schlenk or glovebox techniques, as all Zintl compounds are sensitive to air and moisture. Elements were used as received: K lumps, Acros Organics, 98%; Ga pellets, Alfa Aesar, 99,9999% (metals basis); Bi powder, ChemPur Karlsruhe, 99%. Diphenyl zinc (ZnPh$_2$) was prepared according to a modified literature procedure[54]: a 1:2 mixture of ZnCl$_2$ (0.2 mol/l in THF) and PhMgBr (1.3 mol/l in THF) in dry THF was stirred for 3 h at ambient temperature, before the solvent volume was doubled by addition of dioxane for precipitation of ZnPh$_2$ as colorless crystalline powder. The en and $N,N$-dimethylformamide (DMF; Aldrich, 99.8%) were distilled from CaH$_2$ and stored over 3 Å molecular sieves.

Toluene (Acros Organics, 99%) was distilled from sodium–potassium alloy and stored over 4 Å molecular sieves. Kryptofix® 222 (crypt-222, Merck) was dried under vacuum overnight. A solid with the nominal composition K$_5$Ga$_2$Bi$_4$ was prepared by stoichiometric fusion of the elements in a homogeneous temperature chamber oven. The elements were weighed into a niobium tube that was sealed within an evacuated silica ampule. The mixture was heated to 550 °C, kept at this temperature for 24 h, and then cooled down to room temperature at a cooling rate of 5 K/h, and grinded prior to use.

**Synthesis of [K(crypt-222)]$_6$[K$_2$Zn$_{20}$Bi$_{16}$] (1)**. The starting material with the nominal composition K$_5$Ga$_2$Bi$_4$ (176 mg, 150 µmol), crypt-222 (280 mg, 744 µmol), and ZnPh$_2$ (82 mg, 375 µmol) were combined in a Schlenk tube and dissolved in 4.5 ml of en. After stirring for 2 h, an intense green solution, indicating the formation of bluish-green Bi$_4{}^{2-}$ (ref. [55]), was filtered through densely packed glass wool. The solution was stored at room temperature for one night and layered with 5 ml of toluene the next day. Crystals of **1** suitable for SC-XRD (Supplementary Fig. 1) formed on the wall of the tube after ~10 days. Although the yield per batch is systematically not high (<50 mg), the compound is the only one to crystallize from the reaction mixture besides bluish-green crystals of the known by-product [K(crypt-222)]$_2$Bi$_4$, and thus can be obtained in decent amounts by multiple reactions. While the exact processes during the formation of **1** are not yet clarified and subject to current in-depth studies, we suggest that it takes place under precipitation of elemental Ga and Bi upon the redox process, and under release of Ph$^-$, which may then undergo a potential follow-up reaction with the solvent en to form benzene and (H$_2$NCH$_2$CH$_2$NH)$^-$; such processes have been known for reactions of Zintl anions with metal phenyl compounds[56]. Compound **1** is soluble in dry en and DMF. Although the integrity of the highly charged anion in solution could not be confirmed by means of electrospray ionization mass spectrometry (ESI-MS; which may be a consequence of the corresponding measurement conditions), room temperature solution $^1$H NMR of **1** in DMF-$d^7$ indicate the presence of [K(crypt-222)]$^+$ (Supplementary Fig. 10), hence corroborating the solubility as such.

**Single-crystal X-ray diffraction**. Several data sets for the X-ray structural analyses were collected at $T = 100(2)$ K on different crystals with different sizes and on different diffractometers with Mo–K$_\alpha$ radiation and Cu–K$_\alpha$ radiation, as it was difficult to obtain sufficient data, especially at higher angles. In spite of high absorption, best results were gained from data measured with Cu radiation ($\lambda = 1.54186$ Å) on an area detector system Stoe StadiVari at a GeniX 3D microfocus source. The structure was solved by direct methods (SHELXT)[57]. The refinement was done by full-matrix-least-squares methods against $F^2$ with the program SHELXL[58]. It clearly revealed the $[K_2Zn_{20}Bi_{16}]^{6-}$ anion and the K centers of the [K(crypt)]$^+$ cations (Fig. 1). The large displacement parameters are not the result of an inappropriate absorption correction: in several refinements with different data sets, with different absorption corrections, and even with the uncorrected original data, their size shows only small variations (<20%). As this disorder likely influences the atomic positions of adjacent cations, it is comprehensible that the light atoms of the cryptate ligands could not be localized from the Fourier map. In addition, this explains the intensity decay at higher angles in the data.

**Energy-dispersive X-ray spectroscopy**. The EDS analysis on **1** was performed using the EDS device Bruker XFlash 5010 attached to a JEOL JIB-4601F SEM (implemented in a SEM/focused ion beam dual beam system) operating at 15 kV. Data acquisition was performed with at least 100 s accumulation time. For the analyses, multiple single crystals were tested (Supplementary Figs. 3 and 4, and Supplementary Table 2).

**Quantum chemical calculations**. Structure parameters were optimized with the functional TPSS[59] using basis sets of type dhf-SVP[60] together with corresponding effective core potentials[61] and Coulomb fitting basis sets[62]. The negative charge was compensated with the conductor-like screening model (COSMO)[63], employed with default parameters, except for the cavity radius of Zn. This was set to 2.223 Å, the default value for both neighboring elements Cu and Ga, as well as for K. The induced current density was studied with basis sets of type dhf-TZVP[60]. Scalar-relativistic all-electron calculations were carried out with the diagonal local approximation to the unitary decoupling transformation of the exact two-component (DLU-X2C) Hamiltonian[64–66] and basis sets of type x2c-TZVPall-s[67]. For details, see the Supplementary Discussion. Calculations of $[Hg_8Te_{16}]^{8-}$ were performed with the same settings as above, but employed the same basis set as in ref. [39].

**Powder X-ray diffraction**. Powder X-ray diffraction data were collected on a Stoe StadiMP diffractometer system equipped with a Mythen 1 K silicon strip detector and Cu–K$_\alpha$ radiation ($\lambda = 1.54056$ Å). A sample of the starting material with the nominal composition $K_5Ga_2Bi_4$ was filled into a glass capillary (0.3 mm diameter), which was sealed air-tightly with soft wax. The tube was then mounted onto the goniometer head using wax (horizontal setup) and rotated throughout the measurement.

**Electrospray ionization mass spectrometry**. ESI mass spectra (Supplementary Figs. 8 and 9) were recorded with a Thermo Fischer Scientific Finnigan LTQ-FT spectrometer in the negative ion mode. All samples were prepared inside of a glovebox, where they were dissolved in anhydrous DMF and filtered through teflon syringe filters with a pore size of 0.45 μm. The solutions were injected into the spectrometer with gastight 250 μl Hamilton syringes by syringe pump infusion. All capillaries within the system were washed with dry DMF for 30 min before, and at least 10 min in between measurements to avoid decomposition reactions and consequent clogging.

## Data availability

All data generated or analyzed during this study are included in this published article and its Supplementary information files. The X-ray crystallographic coordinates for the structure reported in this study have been deposited at the Cambridge Crystallographic Data Center (CCDC), under the deposition number 1969162. These data can be obtained free of charge from The CCDCC via www.ccdc.cam.ac.uk/data_request/cif. Input files or sets of all input parameters for TURBOMOLE and GIMIC are available from the corresponding authors upon request. See also the Supplementary Discussion for details on the computational methods noting all non-default parameters for TURBOMOLE. Cartesian coordinates of the optimized structures are listed in the Supplementary Information.

## Code availability

The TURBOMOLE quantum program suite is available from https://www.turbomole.org (Accessed 29 August 2020), and the GIMIC code can be obtained from the GitHub repository at https://github.com/qmcurrents/gimic (Accessed 29 August 2020; open-source, see also ref. [46]). The GitHub repository also includes a sample input for GIMIC. Additionally, the Supplementary Information contains a short note on the use of GIMIC with Python version 2.

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

## Acknowledgements

The authors thank the Deutsche Forschungsgemeinschaft (DFG) for financial support within the framework of GRK 1782. We thank J. L. Vasco, K. Hoffmann, and M. Pyschik for help with the synthesis, Dr. S. Ivlev, M. Marsch, and R. Riedel for help with the diffraction experiments, and M. Hellwig for measuring EDX spectra of 1, and we thank Dr. K. Reiter, Prof. S. Bobev, and Prof. S. C. Sevov for fruitful discussion. Y.J.F. is grateful to Fonds der Chemischen Industrie (FCI) for general support of his Ph.D. studies (Kekulé fellowship) and the German Academic Exchange Service (Deutscher Akademischer Austauschdienst, DAAD) for a fellowship (grant no. 57438025), and Prof. F. Furche for hosting.

## Author contributions

A.R.E. conceived and performed the synthetic experiments, collected single-crystal X-ray crystallographic data, solved and refined the structure for the first time, performed ESI mass spectrometry, and prepared samples for further analyses. W.M. performed sophisticated structure solution and refinement of different data sets, and finalized the structure analysis. Y.J.F., P.B., and F.W. carried out and documented the quantum chemical investigations. S.D. and F.W. supervised the work. All authors co-wrote the paper.

## Funding

## Competing interests

The authors declare no competing interests.
