## [Peer Review File · Nature Communications]

Reviewers' comments:

Reviewer #1 (Remarks to the Author):

This article reports the synthesis and characterisation of an intricate cluster anion composed of K, Zn and Bi centers. The structure is revealed using X-ray crystallography, and the authors show convincingly that no Ga has been incorporated in the product as it is well known that Ga and Zn centers can be mistaken for one another using X-ray crystallography. The anion of the product displays a high degree of symmetry and is unquestionably beautiful. The authors then use computational techniques to probe the electronic properties of the anion. However, I am unconvinced that this work is of extreme importance to scientists in the field. The computational analysis leads to far-fetched conclusions: for a species that shows no experimental evidence of a free Zn₁₂ unit, just analysing this unit in silico is of little benefit to science. Furthermore, linking the rest of the Zn/Bi structure to that of a porphine should be just a cute observation/aside, but is not a claim with any repercussions or substance given that the free entity has not been shown to have an independent existence. Given this statement, I was also less impressed by the claim of aromaticity in the "supporting structure" - computational techniques quantifying aromaticity have been rightfully popular for several years, but the authors apply this to a unit with no independent existence is of little value.

In summary, this result is beautiful that the claims of the importance of the result are over-egged. I do not believe it is suitable for publication in Nature Communications.

Reviewer #2 (Remarks to the Author):

The authors have reported the synthesis and characterization of a large heterometallic Zn/Bi cluster anion, [K₂Zn₂₀Bi₁₆]⁶⁻. I am not well placed to judge the experimental protocols or the characterization techniques but as far as I can judge these have been done very competently. There is no doubt that the structure of the cluster is striking, and the presence of the close-to-zerovalent Zn₁₂ core is a remarkable complement to the (very limited) known structural chemistry of low-valent Zn clusters. The Zn-Zn bond lengths certainly consistent with compounds such as Carmona's Cp₂Zn₂. The authors analysis of the K⁺ binding in terms of the strong electrostatic potential is insightful, as it sheds some light on the balance between 'tight-binding' of cations and their sequestration by macrocyclic ligands that is always an issue in highly anionic clusters such as this.

The bonding analysis is well argued, and makes a convincing case for the localized view of bonding. The separation into an isolated Zn₁₂ unit is also insightful. I am not, however, wholly convinced by the argument that because a [Zn₁₂]⁰ cluster has the global minimum that resembles the crystal structure implies that this unit necessarily carries this charge in the intact structure. The tightly-bound bridging bismuth anions will clearly play an important structural role, and any resemblance between the optimised structure of the Zn₁₂ core and the intact structure may well be coincidental. In particular, I am not sure that the fact that [Zn₁₂]⁶⁺ has a more stable structure indicates that the cluster core is not in an effective +6 state, as the earlier NPA analysis seems to indicate. Presumably the D_{4d}-symmetric

structure shown in Figure S7 is strongly disfavoured by the presence of the bismuth anions (it's not obvious to me how 16 of them might bridge this structure in a symmetric way). A minor point here - what is the meaning of +/-0? Is it not just 0?

The analogy to porphine is an interesting one, and makes a neat connection between inorganic and organic systems. If I have understood correctly, the magnetic calculation was done on the entire cluster, which would be the analogue of a deprotonated porphine ligand with a metal bound to the central cavity, rather than the ligand itself. I don't know whether metal binding has a significant impact on the aromaticity of porphine itself, but this might be a useful comparison to bring into the discussion.

In summary, I find this a very elegant paper based on a beautiful new structure that sheds important new light on low-valent zinc chemistry. There are questions to debate in the electronic structure analysis but there always are in systems of these types, and I think this makes an important contribution. Thus I would recommend publication, subject to the authors considering the points raised above.

Reviewer #3 (Remarks to the Author):

This is another fantastic intermetallic cluster from the Dehnen group which certainly deserves publication. Congratulations! The results are novel and will be of interest to others in the community and the wider field.

Even though the yields are rather low and even though the formation mechanism is unclear (See page 11: "While the exact processes during the formation of 1 are not yet clarified and subject to current in-depth studies") the reaction is reported to be highly reproducible.

Summary: this is a highly interesting molecule and the compound is highly recommended for publication. But due to the very poor single crystal X-ray structure determination allowing only for a "probable formula", some details concerning the composition and charge of the anion should be considered before.

1. In the results section is on one hand the part on the structure description rather short (less than one page plus figure), on the other hand is the theoretical part very detailed and lengthy (approximately 4 pages). More emphasis should be given to the experimental results and the discussion of the crystal structure refinement.

2. The cluster is built from K, Bi and Zn atoms which is found as a part of a highly disordered crystal structure. The disorder inhibits the determination of the exact composition of the studied compound, therefore the formula used in the manuscript is described as "probable formula" in Table S1. The entire crystal units could not be localized, thus other solvate molecules can coordinate to the K atom. In the case of charged ligands such as deprotonated en molecules, which are discussed in the manuscript, the charge of the cluster would differ. This must be considered in the discussion and the calculations.

3. As mentioned in the manuscript play negatively charged Ph ligands as well as deprotonated en molecules a role in this type of solutions. Such Ph ligands could be even attached to the outer cluster atoms. Ligand migration has been observed in other cases. This point should be discussed in the manuscript and also be considered in the theoretical part.

4. The formed clusters dissolve, which is an advantage. In order to see if Ph ligands are present a simple proton NMR spectrum will give the answer. Crypt and [k-crypt]⁺ as well as the other solvent molecules will show up. Since no single C atom is refined in the X-ray data, this is mandatory.

5. The Ph groups are well separated from the other signals. Since the reaction is highly reproducible elemental analysis (minimum C, H, N) could be undertaken. In combination with the NMR spectrum the solvent molecules which might be included in the crystals as solvate molecules could be identified. This is necessary due to the uncertainty of the X-ray data refinement.

6. The ESI mass spectrum is not very helpful since only Bi containing anions are detected. It might be however interesting to compare the spectra with solutions that solely contain the ternary precursor alloy. Do mixed Ga-Bi clusters show up here?

7. The theoretical part is very extensive and the part on the bond orbital description could be shortened. However the calculations should include two points:

i) The HOMO-LUMO gap is given for the assumed isolated subunit: “{Zn₁₂}±0 converges with a reasonable HOMO-LUMO gap of 1.6 eV,” but is not mentioned for 1a. Why? This could be indicative on the charge of the cluster and also if ligands could be attached to the outer atoms.

ii) As pointed out below for the discussion on the crystal structure, the authors should assume also in their calculations four Ga atoms in the centre of the cluster. Such small amount of Ga inside a cluster might not show up in the EDX spectra. Based on the X-ray data Zn and Ga can only hardly be distinguished. Charges indicate that the atoms in the centre might be different. Therefore the differences by substituting the central Zn atoms by Ga should be checked by computational methods (at least changes in the HOMO-LUMO gap) and in the crystal structure refinement (see below).

More details concerning the crystal structure determination should be considered:

C1. Although the crystallographer was doing the best, the result remains inadequate. The shortcomings of the structure determination are addressed in the paper quite clearly:

- crystals are thin needles,
 - nevertheless they show strong absorption,
 - consequently the absorption correction is very difficult,
 - generally the reflection intensities are weak, and additionally decrease rapidly at higher 2 theta angles.
- However, according to CIF and Supp. Inform. the measured crystal was 0.09 mm x 0.12 mm x 0.27 mm, that means not very small. Therefore one should encourage the authors to try a smaller one with the consequence of lower absorption which might even give higher intensities to the weak reflections.

C2. Further

- no single light atom positions of the cryptand molecules can be determined (therefore Squeeze),
- possible content of solvent cannot be determined,
- extraordinarily large displacement ellipsoids of all heavy atoms,

are explained with the supposedly high "dynamics" in the crystal. As mentioned above seems this to be highly speculative with a measuring temperature of 100 K and the given absorption problems. A lower symmetry, which cannot be resolved due to the low intensities, may also play a role. Disregarding the high 2θ angles, $R(\text{int})$ of 18.3% is very high. However disorder of atoms may play a crucial role here and should be discussed.

C3. Another point around the structure determination concerning the composition of the compound is critical. The residual values after the structure refinement might be identical if Zn is replaced by Ga. However, due to the large amount of Zn in the structure at least a slight effect should appear. Despite the EDX data this should be ruled out also by the X-ray data.

C4. As mentioned above: for example, a Ga₄ square in the cluster centre could be discussed because the bonds are the shortest ones at all in the cluster and the displacement parameters of the relevant atoms Zn₅/Zn₆ are smaller than those of the surrounding atoms while they are of the same size if assigned as Ga. This effect is small but present.

Some minor points:

M1. Page 3: „The quoted clusters were obtained by reactions of K₅Bi₄ or [K(crypt-222)]₂(Sn₂Bi₂)-en, respectively, with ZnPh₂“ Since plural “clusters” is used, also Pb containing clusters are included.

M2. Page 3: “ternary solid K₅Ga₂Bi₄“. The term ternary solid is not very clear. Is this here a defined phase of this composition or a mixture with „nominal composition“. In any case a powder diffractogram should be provided in the Supporting Information.

M3. Page 11: The reported assumption on the reaction should be omitted. The coincident formation of Ga(0) and Bi(0) under the assumption that Ga from the ternary solid act as reducing agent is not convincing. There are several other possibilities to make up similar equations.

M4. See Page 11: “metallic by-products, which we could confirm to form in an approximately 1:1 ratio, and which we plan to recycle for the formation of new reactant as they are easily separable from the reaction mixture.” Since the side products appear in such good quantities and due to the fact that the composition of 1 is vague, side products should be characterized.

M5. Page 9: The calculations of sequestering K ions is not very helpful since the reaction was carried out in ethylenediamine which is known to coordinate as well to K cations and can even form complexes in the case of K-18c6 units. In addition none of the crypt units is resolved using the X-ray data. This part should therefore be shifted to Supporting Information.

M6. Atom labelling Bi' in Fig.1 has to be changed: in text p.3, below: distance Bi1-Bi1' 11.4 Å; in Fig.1.a) the distance between the atoms designated as Bi1 and Bi1' is 8.0 Å, while the 11.4-Å distance is between Bi1 atoms at opposite sides.

M7. Supporting Information, Table S1: empirical formula is not Bi₁₆ K₂ Zn₂₀ but Bi₁₆ K₈ Zn₂₀.

M8. Figure caption Fig. S5: in the ESI-MS there are signals of "Bi³⁺" and "Bi₃H⁺". The signal for Bi₃H is of course that at 628 and NOT that at 626.9.

Reviewers' comments:

Reviewer #1 (Remarks to the Author):

This article reports the synthesis and characterisation of an intricate cluster anion composed of K, Zn and Bi centers. The structure is revealed using X-ray crystallography, and the authors show convincingly that no Ga has been incorporated in the product as it is well known that Ga and Zn centers can be mistaken for one another using X-ray crystallography. The anion of the product displays a high degree of symmetry and is unquestionably beautiful. The authors then use computational techniques to probe the electronic properties of the anion. However, I am unconvinced that this work is of extreme importance to scientists in the field. The computational analysis leads to far-fetched conclusions: for a species that shows no experimental evidence of a free Zn_{12} unit, just analysing this unit in silico is of little benefit to science. Furthermore, linking the rest of the Zn/Bi structure to that of a porphine should be just a cute observation/aside, but is not a claim with any repercussions or substance given that the free entity has not been shown to have an independent existence. Given this statement, I was also less impressed by the claim of aromaticity in the “supporting structure” - computational techniques quantifying aromaticity have been rightfully popular for several years, but the authors apply this to a unit with no independent existence is of little value.

Response to the comment: We acknowledge and greatly appreciate this reviewer's critical comments on our work. We took them as an inspiration of how to improve the report. As a response to their critical concerns, we would like to note the following:

(a) *Re: importance to scientists in the field:* we feel that our work contributes to both the study of low valent zinc compounds and to heterometallic clusters in general. Regarding the first, we contribute to the rare class of low valent, metalloid zinc clusters, which are studied for understanding their unique physico-chemical properties as they might be useful as precursors to innovative materials for molecule activation. The latter development is still in its childhood, yet we are eager to provide another class of potential species or precursors to such materials. Regarding the second, our new approach to heterometallic clusters by using the heretofore not applied solid “ $K_5Ga_2Bi_4$ ” opened the door to a new class of both zinc-rich and bismuth-rich clusters that were not accessible via previously described synthetic pathways. We consider this finding very important, as subsequent work already indicated that much more new compounds can be expected to evolve from its use. While the importance of a new finding in many cases is a matter of the viewpoint of a scientist, we hope to be in the position to convince this reviewer of the importance of our finding for the broad field of metal cluster chemistry.

(b) *Re: discussion of the $\{Zn_{12}\}$ unit:* We agree that the modelling of the $\{Zn_{12}\}$ unit is a theoretical “experiment”, although it is not uncommon to fragment a complicated structure and study subunits separately in order to better understand the whole architecture. Hence, we modified and shortened this part – also according to the comments of Reviewer #2 (see below). Still, we would like to add that there exist reports even in highest-ranking journals on purely hypothetical species that did not come along with any experimental results (yet in some cases predicted their appearance), while in our study, all computations are closely related to the experimentally observed cluster compound and just expand on insights into its electronic structure and stabilities of the substructures. In summary, we have the impression that our overall way of approaching the complicated cluster structure is helpful, as can be seen from the comments by the other reviewers.

(c) *Re: discussion of the relationship to porphine:* We understand this reviewer's point, and have therefore emphasized the thought experiment character of this part and the fact that the $\{Zn_8Bi_{16}\}^{9-}$ metallacycle is not known as a separate entity by adding the following line: “In contrast to porphine, the $\{Zn_8Bi_{16}\}^{9-}$ metallacycle is not known as a separate entity.”. As the other reviewers explicitly acknowledged the interesting relationship with known macrocycles and the detection of a weak aromatic character, we decided to apply this change instead of withdrawing the whole section.

(d) *Re: discussion of the aromaticity:* We agree that it was necessary to write this part more carefully. As stated in the previous point, we have indicated the thought experiment character of this whole section by adding a corresponding statement. Nevertheless, we would like to add that our finding has in the meantime even been corroborated by further analyses: the aromaticity was studied by two different approaches, the gauge-including magnetically induced current density (GIMIC) method, for which the

two K^+ ions had to be removed in order to place an integration plane, and by the nucleus-independent chemical shift (NICS) method, which considers the whole $[K_2Zn_{20}Bi_{16}]^{6-}$ cluster. Both methods indicate weak aromaticity (see also the answer to Reviewer #2 below).

In summary, this result is beautiful that the claims of the importance of the result are over-egged. I do not believe it is suitable for publication in Nature Communications.

Response to the comment: We thank the reviewer for giving us the chance to discuss the significance of our work. We do not have the impression that the claims of importance are over-egged in the article. Please note that none of the words “remarkable”, “important”, “interesting”, “importance”, “significance”, or “meaningfulness” appear in our text in the context of our own study, as we tend to let others judge on this. Yet, the other reviewers used this kind of attributes for our findings, which we take as an indication for the correct choice of the journal to which we submitted the manuscript.

Reviewer #2 (Remarks to the Author):

The authors have reported the synthesis and characterization of a large heterometallic Zn/Bi cluster anion, $[K_2Zn_{20}Bi_{16}]^{6-}$. I am not well placed to judge the experimental protocols or the characterization techniques but as far as I can judge these have been done very competently. There is no doubt that the structure of the cluster is striking, and the presence of the close-to-zerovalent Zn_{12} core is a remarkable complement to the (very limited) known structural chemistry of low-valent Zn clusters. The Zn-Zn bond lengths certainly consistent with compounds such as Carmona's Cp_2Zn_2 . The authors analysis of the K^+ binding in terms of the strong electrostatic potential is insightful, as it sheds some light on the balance between 'tight-binding' of cations and their sequestration by macrocyclic ligands that is always an issue in highly anionic clusters such as this.

Response to the comment: Thank you very much for your overall very enthusiastic assessment of our study!

The bonding analysis is well argued, and makes a convincing case for the localized view of bonding. The separation into an isolated Zn_{12} unit is also insightful. I am not, however, wholly convinced by the argument that because a $[Zn_{12}]^0$ cluster has the global minimum that resembles the crystal structure implies that this unit necessarily carries this charge in the intact structure. The tightly-bound bridging bismuth anions will clearly play an important structural role, and any resemblance between the optimised structure of the Zn_{12} core and the intact structure may well be coincidental. In particular, I am not sure that the fact that $[Zn_{12}]^{6+}$ has a more stable structure indicates that the cluster core is not in an effective +6 state, as the earlier NPA analysis seems to indicate. Presumably the D_{4d} -symmetric structure shown in Figure S7 is strongly disfavoured by the presence of the bismuth anions (it's not obvious to me how 16 of them might bridge this structure in a symmetric way). A minor point here - what is the meaning of +/-0? Is it not just 0?

Response to the comment: Indeed, the stability of the isolated Zn_{12} unit for $q = 0$ should be seen rather as an insightful additional fact than as an argument; nevertheless, all calculations indicate that this unit is clearly low-valent (although not zero-valent). To avoid any over-interpretation, we significantly shortened this part, which now reads as follows: “A natural population analysis (NPA)^[36] performed on the optimized structure of **1a** (Fig. 2b) yields charges of +0.05 for the inner four Zn atoms and +0.71 for the other eight Zn atoms. The latter is a typical value for an oxidation state of +I (compare, for instance, $ZnCl$: +0.71, $ZnBr$: +0.67), so this unit as a whole is clearly low-valent. We note in passing that this $\{Zn_{12}\}$ unit as an isolated species is stable even as a neutral species. Structure optimizations carried out for charges $q = 0, +2, +6$ (Fig. S7) show that the structure of the $\{Zn_{12}\}$ moiety persists in an isolated form, for $q = 0$ even with a reasonable HOMO-LUMO gap of 1.6 eV, indicating the general stability of this cluster unit. However, a second local minimum was found that is less stable for $\{Zn_{12}\}^{\pm 0}$ (+57 kJ/mol) and $\{Zn_{12}\}^{2+}$ (+9 kJ/mol), but is favorable for $\{Zn_{12}\}^{6+}$ (-138 kJ/mol). In summary, the calculations confirm **1a** as being a cluster with the largest low-valent subunit of group 10 metals known to date.”

Regarding the minor point: of course, ± 0 is equal to just 0. We tend to write ± 0 throughout to avoid any confusion with a superscript letter “O”, which is used occasionally in our work to indicate the presence of O-donor ligands (not in this work, though).

The analogy to porphine is an interesting one, and makes a neat connection between inorganic and organic systems. If I have understood correctly, the magnetic calculation was done on the entire cluster,

which would be the analogue of a deprotonated porphine ligand with a metal bound to the central cavity, rather than the ligand itself. I don't know whether metal binding has a significant impact on the aromaticity of porphine itself, but this might be a useful comparison to bring into the discussion.

Response to the comment: We thank the reviewer for bringing this up, which prompted us to extend the study and also the corresponding section in the manuscript. Two different approaches are now considered to study the degree of aromaticity: first, the gauge-including magnetically induced current density (GIMIC) method, for which the two K^+ had to be removed in order to place an integration plane and thus to study $[Zn_{20}Bi_{16}]^{8-}$, and second, the nucleus-independent chemical shift (NICS) approach for the cluster both with and without the K^+ ions – with very similar values of -3.6 ppm and -4.4 ppm, respectively. Additionally, we have included studies on Zn(II) porphyrin in the revised version of our report. Porphine and Zn(II) porphyrin show the same current strength and the same degree of aromaticity. In the revised version of our work, current strengths are therefore given for $[Zn_{20}Bi_{16}]^{8-}$, benzene, porphine, zinc porphyrine and $[Hg_8Te_{16}]^{8-}$, with the latter being topologically very similar to porphyrin but non-aromatic. NICS values are given for $[Zn_{20}Bi_{16}]^{8-}$, $[K_2Zn_{20}Bi_{16}]^{6-}$, benzene, porphine and $[Hg_8Te_{16}]^{8-}$. Note that NICS values could not be calculated for Zn(II) porphyrine, as this method requires a ghost atoms to be placed at the center of mass, and thus at the same position as the zinc atom (which has a local ring current itself).

In summary, I find this a very elegant paper based on a beautiful new structure that sheds important new light on low-valent zinc chemistry. There are questions to debate in the electronic structure analysis but there always are in systems of these types, and I think this makes an important contribution. Thus I would recommend publication, subject to the authors considering the points raised above.

Response to the comment: Thank you very much for your justified and helpful comments, all of which were taken into consideration during the revision of our work.

Reviewer #3 (Remarks to the Author):

This is another fantastic intermetallic cluster from the Dehnen group which certainly deserves publication. Congratulations! The results are novel and will be of interest to others in the community and the wider field. Even though the yields are rather low and even though the formation mechanism is unclear (See page 11: “While the exact processes during the formation of 1 are not yet clarified and subject to current in-depth studies”) the reaction is reported to be highly reproducible.

Summary: this is a highly interesting molecule and the compound is highly recommended for publication. But due to the very poor single crystal X-ray structure determination allowing only for a “probable formula”, some details concerning the composition and charge of the anion should be considered before.

Response to the comment: Thank you very much for your overall very enthusiastic assessment of your study! We were pleased to use your helpful comments for a thorough revision of the manuscript and the supplement.

1. In the results section is on one hand the part on the structure description rather short (less than one page plus figure), on the other hand is the theoretical part very detailed and lengthy (approximate 4 pages). More emphasis should be given to the experimental results and the discussion of the crystal structure refinement.

Response to the comment: We agree with this point. The reason why we kept this part relatively short in the original submission, while it was outlined more detailed in the Supplementary Information, was that we anticipated that the theoretical part required some more explanations – so it obtained priority in the light of the restricted maximum number of words. Yet, in the meantime, we learned that we can use more words than previously anticipated, which we happily used to significantly extend the synthesis and structure description and discussion section. Some shortenings of the theoretical part (see the answers to Reviewer #1 above) additionally helped to present a better-balanced report now.

2. The cluster is built from K, Bi and Zn atoms which is found as a part of a highly disordered crystal structure. The disorder inhibits the determination of the exact composition of the studied compound,

therefore the formula used in the manuscript is described as “probable formula” in Table S1. The entire crypt units could not be localized, thus other solvate molecules can coordinate to the K atom. In the case of charged ligands such as deprotonated en molecules, which are discussed in the manuscript, the charge of the cluster would differ. This must be considered in the discussion and the calculations.

Response to the comment: Just to avoid any misunderstanding: the cluster structure itself is not disordered in a sense that would inhibit the unambiguous assignment of atom types (in contrast to many heterometallic clusters with spherical shape that show significant orientational disorder); we just observe several positions of each of the metal atoms in very close proximity, which we put down on slightly different, statistically distributed orientations in the crystal. This is explained in more detail now. The disorder of the cryptand molecules, in contrast, is much more distinct, which is generally a known phenomenon, but arises to an extent in this case which inhibits the localization of the atoms. We explain this fact much more detailed now, as well. Of course, we also checked the data set for the presence of other possible solvate molecules (like en) or cations (like deprotonated en). However, there is no evidence whatsoever for their allocation, while the spatial distribution of six K^+ cations within the crystal structure of **1** makes perfect sense and does not produce unreasonable voids (see Supplementary Figure 2).

Regarding the charge of the cluster: fortunately, DFT calculations are very strong in determining the correct charge of a cluster. The MO occupancy in the frontier orbital region, and the resulting size of the HOMO-LUMO gap, are very strong indicators for the electron number (hence, also the total charge) to be wrong or correct. In our case, the cluster’s electronic structure would significantly suffer from any electrons to be added or withdrawn.

In summary, we understand that our description of the crystal structure was definitely too brief. Therefore, a significantly more extended discussion of these structural issues has been added to both the Results and the Methods section of the manuscript and the Supplementary Discussion section of the Supplementary Information file.

3. As mentioned in the manuscript play negatively charged Ph ligands as well as deprotonated en molecules a role in this type of solutions. Such Ph ligands could be even attached to the outer cluster atoms. Ligand migration has been observed in other cases. This point should be discussed in the manuscript and also be considered in the theoretical part.

Response to the comment: Thank you for this interesting input. We have double-checked the data set for this. However, while the disordered cryptand molecules produce corresponding (smeared) electron density around the K^+ cations, there is no electron density found in proximity of the cluster atoms, which would allow to define or model Ph groups in this region. We take all of these findings as sufficiently strong indications for the Zn atoms to be clearly “naked” in **1**. A corresponding statement was added to the manuscript.

4. The formed clusters dissolve, which is an advantage. In order to see if Ph ligands are present a simple proton NMR spectrum will give the answer. Crypt and [k-crypt]⁺ as well as the other solvent molecules will show up. Since no single C atom is refined in the X-ray data, this is mandatory.

Response to the comment: We thank the reviewer for this valuable suggestion, which served to clarify several issues. According to this suggestion, a proton NMR spectrum of crystals of **1** was recorded in DMF-d7 (shown in new **Supplementary Figure 10**). The spectrum is dominated by the signals of crypt-222 (from the $[\text{K}(\text{crypt-222})]^+$ counterions). Signals of smaller intensity indicate the presence of solvents en (broad signal of NH_2 at 1.5 ppm and sharp signal of the methylene groups at 2.5 ppm) and toluene (two signals around 7 ppm, see inset). The intensity of these signals is much smaller than the intensity of the signals of the cryptand. Hence we attribute these signals to (residual) crystal solvent in **1** (upon drying the crystals *in vacuo* before re-dissolving them in the NMR solvent), or to residues on the crystal surfaces even upon drying (as we cannot quantify the crystal solvent content this way, it is not considered in the formulas provided in **Supplementary Table 1**). However, we can conclude with certainty that no Ph groups are attached to the Zn atoms of the $[\text{K}_2\text{Zn}_{20}\text{Bi}_{16}]^{6-}$ cluster: such substituents, which would be present as 8 or 12 or 20 equivalents per formula unit owing to the cluster's symmetry, would not be removed by the drying procedure. Hence, they would appear in an 8:6 or 12:6 or 20:6 ratio with regard to the 6 crypt-222 molecules per formula unit, which is not at all reflected by the signal intensities. We can also conclude that the amount of en detected by NMR spectroscopy would not even be enough to add up to one single additional (deprotonated en) counterion – in accordance with our statement on the composition and charge above.

5. The Ph groups are well separated from the other signals. Since the reaction is highly reproducible elemental analysis (minimum C, H, N) could be undertaken. In combination with the NMR spectrum the solvent molecules which might be included in the crystals as solvate molecules could be identified. This is necessary due to the uncertainty of the X-ray data refinement.

Response to the comment: We agree that a CHN analysis of **1** would provide even more information in this regard. However, owing to the simultaneous precipitation of metal powder, and the co-crystallization of the by-product $[\text{K}(\text{crypt-222})]\text{Bi}_4$, and owing to the very fine habitus of the crystals, it is nearly impossible to collect pure material of **1** in a sufficiently large amount. We therefore hope for this reviewer's understanding and acceptance of the explanations given above based on the new NMR data.

6. The ESI mass spectrum is not very helpful since only Bi containing anions are detected. It might be however interesting to compare the spectra with solutions that solely contain the ternary precursor alloy. Do mixed Ga-Bi clusters show up here?

Response to the comment: As a general observation, ESI MS data for the binary anions $(\text{TrBi}_3)^{2-}$ falls short. According to literature, and to our own (numerous) studies, no binary (Ga_xBi_y) fragment was ever observed in ESI MS experiments. ESI mass spectra of the deep green extraction solutions of “ $\text{K}_5\text{Ga}_2\text{Bi}_4$ ” generally look like those shown in **Supplementary Figure 8** and **Supplementary Figure 9**. Slow evaporation of such extraction solutions (yielding crystalline $[\text{K}(\text{crypt-222})]_2(\text{GaBi}_3)\cdot\text{en}$), re-dissolving of the solid in dmf, and subsequent ESI MS measurements cause fewer Bi_x fragments to show up, yet until today, such studies have never produced signals of any (Ga_xBi_y) fragments. We therefore assume that these binary species are too sensitive as being transferrable into the gas phase, which is in agreement with their overall tendency to react under release of elemental Ga.

7. The theoretical part is very extensive and the part on the bond orbital description could be shortened. However the calculations should include two points:

- i) The Homo-LUMO gap is given for the assumed isolated subunit: “ $\{\text{Zn}_{12}\} \pm 0$ converges with a reasonable HOMO-LUMO gap of 1.6 eV,” but is not mentioned for **1a**. Why? This could be indicative on the charge of the cluster and also if ligands could be attached to the outer atoms.
- ii) As pointed out below for the discussion on the crystal structure, the authors should assume also in their calculations four Ga atoms in the centre of the cluster. Such small amount of Ga inside a cluster might not show up in the EDX spectra. Based on the X-ray data Zn and Ga can only hardly be distinguished. Charges indicate that the atoms in the centre might be different. Therefore the differences by substituting the central Zn atoms by Ga should be checked by computational methods (at least changes in the HOMO-LUMO gap) and in the crystal structure refinement (see below).

Response to the comment: As mentioned above, the lengths of the experimental and the theory part are much better balanced in the revised version of the manuscript in accordance with this comment. Furthermore, we agree that more substance is needed to convince the reader about the given composition. Regarding the points above, we respond as follows:

Ad i) We apologize for the accidental omission of this information in the original submission. The value of the gap was added to the revised version of the manuscript in the beginning of the quantum chemistry section which now reads: “We optimized^[33] the geometric structure at DFT level (TPSS/dhf-TZVP). The calculated molecular orbitals show a significant HOMO-LUMO gap of 1.4 eV, but no further significant gaps in the region of frontier orbitals which indicates that the total electron number and thus the assumed composition are correct. For a plausible assignment of bonds in $[\text{K}_2\text{Zn}_{20}\text{Bi}_{16}]^{6-}$, we carried out a Boys localization...”.

Ad ii) This is a very important comment. We have addressed this point into detail and performed both the X-ray refinement and the DFT calculations under consideration of an inner Ga_4 ring instead of an inner Zn_4 ring. The new results unambiguously prove our assumption to be correct (see also our responses to points C3 and C4 below): First, the refinement of the X-ray data lead to worse results. Second, the frontier orbital region resulting from the DFT calculations clearly indicate that the cluster would not stand an uptake of four additional electrons (such as introduced by a replacement of 4 Zn with 4 Ga atoms, see above). Yet, to finally rule out the presence of a Ga_4 ring in the cluster center instead of a Zn_4 ring, we calculated a corresponding cluster anion “ $[\text{K}_2\text{Ga}_4\text{Zn}_{16}\text{Bi}_{16}]^{6-}$ ”, which exhibits the following frontier orbital situation:

MO no	irreducible representation	occupation (no of e ⁻)	Energy
383.	127e		-0.067081 H = -1.825 eV
382.	76b2	2.000	-0.075267 H = -2.048 eV
381.	78a1	2.000	-0.077739 H = -2.115 eV
380.	126e	4.000	-0.120678 H = -3.284 eV

The four additional electrons that are introduced by replacing four Zn atoms with four Ga atoms lead to an occupation of the former LUMO and LUMO+1 (irreps 78a1 and 76b2). Accordingly, they are 1.2 eV higher in energy than that of the former HOMO level (irrep 126e). Moreover, the new HOMO-LUMO gap in this hypothetical cluster is unreasonably small, 0.2 eV. These results clearly show that the cluster is not prone to possessing four additional electrons, which is a very strong indication for the inner atoms to be Zn. In other words, the four inner atoms could only be considered as Ga atoms, if the cluster’s overall charge would be reduced to 2⁻ (instead of 6⁻), hence “ $[\text{K}_2\text{Zn}_{16}\text{Ga}_4\text{Bi}_{16}]^{2-}$ ” (instead of isoelectronic $[\text{K}_2\text{Zn}_{20}\text{Bi}_{16}]^{6-}$). This is clearly ruled out by the detection of six additional K^+ counterions in the crystal structure of **1** and the corresponding electron density representing six disordered cryptand molecules in the voids between the anions. Furthermore, our long-standing experience tells us that clusters of this size would not crystallize along with 2 counterions only. We have added this consideration to the revised manuscript as “an isoelectronic alternative including Ga atoms in the cluster center would require a lower overall charge, hence “ $[\text{K}_2\text{Zn}_{16}\text{Ga}_4\text{Bi}_{16}]^{2-}$ ”, which is clearly ruled out by the detection of six additional K^+ counterions in the crystal structure of **1** and the corresponding electron density representing six disordered cryptand molecules in the voids between the anions” – to be found at the outset of the Quantum Chemical Calculations part.

Finally, we note that according to our experience, even small amounts of Ga are well-detectable by EDX spectra, see for instance the EDX data for $[\text{K}(\text{crypt-222})]_3[\text{Sm}@\text{Ga}_2\text{HBi}_{11}]_{0.9}[\text{Sm}@\text{Ga}_3\text{H}_3\text{Bi}_{10}]_{0.1}\cdot\text{en}\cdot\text{tol}$, provided in the Supporting Information of *Angew. Chem. Int. Ed.* **2014**, *53*, 11979–11983. These indicated the presence of 2.26 atom% besides 10.28 atom% of Bi, which was an excellent match with the theoretical values of 2.08 atom% and 10.92 atom%, respectively. Note that in these measurements, the amount of Ga was even slightly overestimated.

Hence, we assume that the accuracy of the EDX spectrum of compound **1**, which indicates no Ga contents at all (0.0%, see **Supplementary Figure 4** and **Supplementary Table 2**), allows us to conclude that the compound does not include any Ga. Still, we acknowledge that this reviewer has prompted us to clarify this important question.

More details concerning the crystal structure determination should be considered:

C1. Although the crystallographer was doing the best, the result remains inadequate. The shortcomings of the structure determination are addressed in the paper quite clearly:

- crystals are thin needles,
 - nevertheless they show strong absorption,
 - consequently the absorption correction is very difficult,
 - generally the reflection intensities are weak, and additionally decrease rapidly at higher 2θ angles.
- However, according to CIF and Supp. Inform. the measured crystal was 0.09 mm x 0.12 mm x 0.27 mm, that means not very small. Therefore one should encourage the authors to try a smaller one with the consequence of lower absorption which might even give higher intensities to the weak reflections.

Response to the comment: Indeed, several crystals, also of smaller size, were investigated, as usual, and as stated in the Methods section of the revised main document. The problems are not so much due to weak intensities or absorption problems (which are responsible for the poor $R(\text{int})$ values) but due to the disorder of the cryptand molecules. This has been addressed more clearly in the Results and Methods sections of the revised manuscript and the Supplementary Discussion section of the revised Supplementary Information.

C2. Further

- no single light atom positions of the cryptand molecules can be determined (therefore Squeeze),
 - possible content of solvent cannot be determined,
 - extraordinarily large displacement ellipsoids of all heavy atoms,
- are explained with the supposedly high "dynamics" in the crystal. As mentioned above seems this to be highly speculative with a measuring temperature of 100 K and the given absorption problems. A lower symmetry, which cannot be resolved due to the low intensities, may also play a role. Disregarding the high 2θ angles, $R(\text{int})$ of 18.3% is very high. However disorder of atoms may play a crucial role here and should be discussed.

Response to the comment: We thank the reviewer for pointing toward these issues. The term "dynamics" actually is misleading; it was thus replaced by "orientational disorder" in the revised manuscript and Supplementary Information, and the features were explained in more detail; as mentioned above, the high $R(\text{int})$ indeed is a consequence of the very high absorption. Possible lower symmetry had been checked, which did not lead to a change of our original assumption. Corresponding comments on both issues were added to the revised documents.

C3. Another point around the structure determination concerning the composition of the compound is critical. The residual values after the structure refinement might be identical if Zn is replaced by Ga. However, due to the large amount of Zn in the structure at least a slight effect should appear. Despite the EDX data this should be ruled out also by the X-ray data.

Response to the comment: Please, see our response to issue C4 below, and our response to comment 7ii above.

C4. As mentioned above: for example, a Ga₄ square in the cluster centre could be discussed because the bonds are the shortest ones at all in the cluster and the displacement parameters of the relevant atoms Zn₅/Zn₆ are smaller than those of the surrounding atoms while they are of the same size if assigned as Ga. This effect is small but present.

Response to the comment: As discussed and explained in detail above, our assumption of the absence of any Ga atoms in the cluster core is based on the following arguments (please, see our response to comment 7ii above, too):

1. Refinement of the inner square of Zn atoms (Zn₅, Zn₆ and equivalents by 2-fold axis) as Ga leads to slightly higher $wR2$ and $R1$ values (Zn: 21.62/7.32%, Ga: 21.63/7.33%).

2. The slightly smaller displacement ellipsoids of Zn5 and Zn6 as compared with the surrounding Zn atoms may be explained by the common observation that the vibrations are smaller in the center of a cluster than in its periphery. When refined as Ga atoms, the mean square displacements rise by about 0.006 Å only.

3. Replacement of Zn5 and Zn6 with Ga would have been detected in the EDX spectra: four atoms represent 10.5 atom%; as we estimate the accuracy of this measurement to be about $\pm 2\%$, we would have detected a signal. In contrast, the spectrum recorded (see **Supplementary Figure 4** and **Supplementary Table 2**) indicates a perfect absence of Ga (0.0%).

4. The results of DFT calculations of a hypothetical anion “[K₂Ga₄Zn₁₆Bi₁₆]⁶⁻” strongly support the correctness of our original assumption (see detailed explanation above).

Some minor points:

M1. Page 3: „The quoted clusters were obtained by reactions of K₅Bi₄ or [K(crypt-222)]₂(Sn₂Bi₂)·en, respectively, with ZnPh₂“ Since plural “clusters” is used, also Pb containing clusters are included.

Response to the comment: Thank you very much for your comment. The entry was corrected to read as follows now: “The quoted clusters were obtained by reactions of K₅Bi₄ or [K(crypt-222)]₂(Tt₂Bi₂)·en, respectively, with ZnPh₂ (crypt-222 = 4,7,13,16,21,24-hexaoxa-1,10-diazabicyclo[8.8.8]hexacosan; Tt = Sn, Pb; en = ethane-1,2-diamine)”.

M2. Page 3: “ternary solid K₅Ga₂Bi₄“. The term ternary solid is not very clear. Is this here a defined phase of this composition or a mixture with „nominal composition“. In any case a powder diffractogram should be provided in the Supporting Information.

Response to the comment: So far, it was not possible to determine the exact identity of the starting material. This was also confirmed by Slavi C. Sevov and his former PhD student Svilen Bobev, who we contacted and who were so kind to share their experiences with us during preparation of this revision (see addition to the acknowledgement). We also applied an optimized synthetic protocol according to the recommendation by Svilen Bobev, which involves lower temperatures (see changes in the Methods Section). In the revised Supplementary Information, we provide the powder X-ray diffraction diagram (**Supplementary Figure 11**), as requested, which clearly indicates (a) a very poor crystallinity of the solid, despite very slow cooling during its preparation in a chamber oven (see details added to the Methods Section in the main document), and (b) no match with any known solid comprising the elements K, Ga, and/or Bi (see the new chapter in the **Supplementary Discussion**).

Such being the case, we cannot provide more information on the precursor, beside naming its nominal composition and the fact that only this “mixture” serves to (reproducibly!) form the title compound reported in our manuscript. Hence, we agree that the term “ternary solid” may be inappropriate. We thus changed it to »a ternary mixture of the nominal composition “K₅Ga₂Bi₄”« (long version) or »“K₅Ga₂Bi₄”« (in quotation marks, short version) throughout the manuscript in accordance with this reviewer’s suggestion.

M3. Page 11: The reported assumption on the reaction should be omitted. The coincident formation of Ga(0) and Bi(0) under the assumption that Ga from the ternary solid act as reducing agent is not convincing. There are several other possibilities to make up similar equations.

Response to the comment: We understand this reviewer’s concern, and agree that the reaction schemes only represent “plausible suggestions” (as was stated in the manuscript). We therefore decided to move the suggested reaction schemes to the Supplementary Information. However, it might be interesting to this reviewer, that we managed in the meantime to crystallize [K(crypt-222)]₂Bi₄ from the reproduction reactions mixtures, and also prove the presence of crystalline Bi metal in the residue (see also response to comment M4 below); hence these parts of the suggested reaction scheme were corroborated at least.

M4. See Page 11: “metallic by-products, which we could confirm to form in an approximately 1:1 ratio, and which we plan to recycle for the formation of new reactant as they are easily separable from the reaction mixture.” Since the side products appear in such good quantities and due to the fact that the composition of 1 is vague, side products should be characterized.

Response to the comment: We reproduced the reaction and investigated the residue upon filtration of the mother liquor from which compound **1** crystallizes upon layering. The residue was investigated by means of μ -XFS and powder-X-ray diffraction. According to μ -XFS, the reaction does not seem to be quantitative according to the reaction scheme shown previously in Equation (5), as we detect Zn in the residue, see the following figure and table (note that the presence Nb metal is an (unreactive) leftover from the production of the starting material in an Nb ampoule). We cannot exclude that compound **1** precipitated prior to layering (in an amorphous manner; not detectable in PXRD), thereby falsifying the element ratio. Although both metals clearly form (in a approximately Ga:Bi = 1:2 ratio), the result of this new measurement prompted us to follow the reviewer’s comment above and refrain from a speculation about the reaction scheme in the main document.

Map Date:15.04.2020 15:01:24 ImpD.:111kcps

El	Z Series	unn. C [Mass%]	norm. C [Mass%]	Atom C [Atom%]	Error (1 Sigma) [Mass%]
K	19 K-Series	5,46	19,62	45,72	0,00
Bi	83 L-Series	15,86	57,00	24,84	0,00
Zn	30 K-Series	2,77	9,96	13,87	0,00
Ga	31 K-Series	2,06	7,40	9,67	0,00
Nb	41 K-Series	1,67	6,02	5,90	0,00
Rh	45 K-Series	0,00	0,00	0,00	0,00
Sum:		27,82	100,00	100,00	

The PXRD measurement, however, indicated that Bi crystallizes, as suggested. Indeed, Bi is the only crystalline compound to appear in the precipitate of (mainly) by-products, as indicated by the powder diffraction diagram shown below (see the diffraction pattern of α -Ga in the inset, with no match).

M5. Page 9: The calculations of sequestering K ions is not very helpful since the reaction was carried out in ethylenediamine which is known to coordinate as well to K cations and can even form complexes in the case of K-18c6 units. In addition none of the crypt units is resolved using the X-ray data. This part should therefore be shifted to Supporting Information.

Response to the comment: The coordination of K^+ by other agents than crypt-222 or 18-crown-6, hence the solvent en or the anions themselves, was reported in a few cases, yet, this is still a relatively rare finding. For this reason, we decided to keep this very brief paragraph in the main document. However, we added a sentence in order to refer to all reported cases in the revised version of the manuscript. This includes the case, in which a complexation of K^+ by en and 18-crown-6 coincides.

M6. Atom labelling Bi' in Fig.1 has to be changed: in text p.3, below: distance Bi1-Bi1' 11.4 Å; in Fig.1.a) the distance between the atoms designated as Bi1 and Bi1' is 8.0 Å, while the 11.4-Å distance is between Bi1 atoms at opposite sides.

Response to the comment: Thank you very much for your comment. Figure 1 was revised, and the labelling scheme in the figure and the text were adopted to match with each other.

M7. Supporting Information, Table S1: empirical formula is not Bi₁₆ K₂ Zn₂₀ but Bi₁₆ K₈ Zn₂₀.

Response to the comment: Thank you very much for pointing to this error, which has been corrected.

M8. Figure caption Fig. S5: in the ESI-MS there are signals of "Bi³⁺" and "Bi₃H⁺". The signal for Bi₃H⁺ is of course that at 628 and NOT that at 626.9.

Response to the comment: Thank you for pointing out that mistake. It has been corrected.

We would like to thank the Editor and all Reviewers for their thorough inspection of our work and their helpful suggestions and comments!

REVIEWER COMMENTS

Reviewer #2 (Remarks to the Author):

The authors have addressed the majority of the concerns that I raised in the original review and also, as far as I can judge, those of the other reviewers. Whilst some of the comparisons drawn (to porphine, for example), are undoubtedly speculative, I think it is incumbent on authors to establish links with other areas of chemistry, even when they are perhaps not obvious. Overall, I would support the publication of this work in Nature Comm.

Reviewer #3 (Remarks to the Author):

The manuscript has improved and I appreciate the efforts of the authors especially concerning the theoretical part. However the major concern with respect to the chemical composition of compound 1 could not be ruled out.

The authors still declare in the manuscript the formation of „ thin, black crystals of [K(crypt-222)]₆[K₂Zn₂₀Bi₁₆]“, even though in the Supplementary Information they inform that this is a “probable formula” (Supplementary Table 1). There is no indication in the manuscript that virtually none of the crypt units is verified from the crystal structure analysis. The formula given in Supplementary Table 1 is “C₁₄₄ H₂₈₈ Bi₁₆ K₈ N₁₆ O₄₈ Zn₂₀” that corresponds to “eight” and not “six” crypt units as used in the manuscript.

Further Supplementary Fig. 2 still suggests that the crystal packing is known, however it represents solely a “modeled section” of 1 with completed surroundings. Six (or eight?) independent crypt molecules corresponds to 372(!) or even more independent atoms that are missing in the single crystal structure refinement. The now added sentence “the light atoms of the cryptate ligands could not be localized from the Fourier map” is strongly misleading since no atom of the crypt unit could be localized.

The authors state in the revised version that [K(crypt-222)]Bi₄ forms as a byproduct. The now presented NMR spectrum can therefore also originate from the co-crystallized byproduct [K(crypt-222)]Bi₄. Enclosures of crystals in salts that co-crystallize are frequent. In addition the formula is [K(crypt-222)]Bi₄ irritating since it corresponds to a paramagnetic compound. Also concerning the second product of the reaction. It is not clear, how the composition of [K(crypt-222)]Bi₄ was determined.

In summary, the manuscript represents the description of an interesting anion and especially the theoretical work is well performed. However the experimental data do not allow the conclusions drawn.

Therefore I cannot recommend publication as long as the crystal structure determination has not improved. The authors should be encouraged to try other counter ions for crystallization or using single-crystal synchrotron data.

Reviewers' comments:

Reviewer #2 (Remarks to the Author):

The authors have addressed the majority of the concerns that I raised in the original review and also, as far as I can judge, those of the other reviewers. Whilst some of the comparisons drawn (to porphine, for example), are undoubtedly speculative, I think it is incumbent on authors to establish links with other areas of chemistry, even when they are perhaps not obvious. Overall, I would support the publication of this work in Nature Comm.

Response to the comment: Thank you very much for this very positive assessment!

Reviewer #3 (Remarks to the Author):

The manuscript has improved and I appreciate the efforts of the authors especially concerning the theoretical part. However the major concern with respect to the chemical composition of compound 1 could not be ruled out.

The authors still declare in the manuscript the formation of „, thin, black crystals of [K(crypt-222)]₆[K₂Zn₂₀Bi₁₆]“, even though in the Supplementary Information they inform that this is a “probable formula” (Supplementary Table 1). There is no indication in the manuscript that virtually none of the crypt units is verified from the crystal structure analysis. The formula given in Supplementary Table 1 is “C₁₄₄ H₂₈₈ Bi₁₆ K₈ N₁₆ O₄₈ Zn₂₀” that corresponds to “eight” and not “six” crypt units as used in the manuscript.

Response to the comment: Actually, we did clearly indicate that no crypt molecule was refined in the revised manuscript – see the text that was added to page 4 of the previous manuscript version: “Unusually high, yet reasonable, displacement parameters in the structure of **1a** point to orientational disorder of the metal atoms over close positions, which also hampers the localization of the atomic positions of the light atoms of the cryptate ligands from the Fourier map. Yet, the spatial demand around the K atoms (see **Supplementary Figure 2**) and the features of the electron density distribution in the Fourier maps agrees with the assumption of [K(crypt-222)]⁺ cations.”

However, to point to this intrinsic issue even more clearly, we have modified the first sentence of the Results chapter (on page 3) to read as follows now: “The reaction of “K₅Ga₂Bi₄” with ZnPh₂ in a 1:2.5 molar ratio in en/crypt-222 at room temperature, subsequent filtration and layering of the solution with toluene affords thin, black crystals of **the most probable composition** [K(crypt-222)]₆[K₂Zn₂₀Bi₁₆] (**1**; see **Supplementary Figure 1**).”

We apologize for the error in the formula in Supplementary Table 1, and we thank this reviewer for pointing to it! The formula was corrected, and also the formula weight and the absorption coefficient, both of which are dependent on this number. We have uploaded a corrected version of the CIF (both to the Editorial Office and to CCDC).

Further Supplementary Fig. 2 still suggests that the crystal packing is known, however it represents solely a “modeled section” of 1 with completed surroundings. Six (or eight?) independent crypt molecules corresponds to 372(!) or even more independent atoms that are missing in the single crystal structure refinement. The now added sentence “the light atoms of the cryptate ligands could not be localized from the Fourier map” is strongly misleading since no atom of the crypt unit could be localized.

Response to the comment: Although we thought that the term “modeled section” would exactly point to the fact that the crystal structure was determined by modelling the electron density of the crypt ligands, we agree that some more explanation should be added to the Figure caption of Supplementary Figure 2. We have modified it to start out as follows now: “**Supplementary Fig. 2 | Two unit cells of the**

modeled section of the crystal structure of 1. Yellow, semi-transparent spheres with a 5 Å diameter are drawn at the positions of well-refined atoms K3 and K4 to indicate the approximate spatial demand of the [K(crypt-222)]⁺ cations, the crypt ligands of which could not be localized from the Fourier map (see **Supplementary Discussion**); the size of the sphere was derived from known crystal structures of other compounds with [K(crypt-222)]⁺ cations.”

The term “light atoms ...” was used for C, N, O, and H atoms, to point to the contrast with the much heavier K, Zn and Bi atoms of the anion and the cations without ligands. However, we have re-phrased the sentence to avoid any misunderstandings: “Unusually high, yet reasonable, displacement parameters in the structure of **1a** point to orientational disorder of the metal atoms over close positions. Obviously, this also involves disorder of the cations, which in the presence of the heavy atoms of the anions hampers the localization of the atomic positions of the light atoms of the cryptate ligands from the Fourier map.”

Regarding the comment about the missing atoms, we would like to bring the following to this reviewer’s attention: The asymmetric unit of compound **1** comprises 39 C, N, and O atoms from the crypt ligands, which are missing in the refinement model, thus we are missing 156 of these “light atoms” per K₆[K₂Bi₁₆Zn₂₀] formula unit. However, the well-localized 44 heavy atoms (11 per asymmetric unit) contribute 62.7% of the total number of electrons (including H atoms). Thus, we think that the application of the back-Fourier-transform method is the adequate way of treating this structure, and moreover, the partial structure of the cluster anion is well-defined and correct.

The authors state in the revised version that [K(crypt-222)]Bi₄ forms as a byproduct. The now presented NMR spectrum can therefore also originate from the co-crystallized byproduct [K(crypt-222)]Bi₄. Enclosures of crystals in salts that co-crystallize are frequent. In addition the formula is [K(crypt-222)]Bi₄ irritating since it corresponds to a paramagnetic compound. Also concerning the second product of the reaction. It is not clear, how the composition of [K(crypt-222)]Bi₄ was determined.

Response to the comment: As said in the caption, “**Supplementary Fig. 10 | ¹H-NMR spectrum of a solution of crystals of 1 in DMF-d₇.**”, the spectrum was taken from single crystals of the product, which can be easily distinguished by shape and color from the greenish-blue crystals of the by-product (at least in an amount suitable for NMR studies).

The formula of the by-product contained a typo: it is actually [K(crypt-222)]₂Bi₄, of course (see the reference), and as such, not paramagnetic. Thank you very much for pointing to this error, which was corrected in the revised version of the manuscript. The compound was identified by its known crystal color and habitus, which was verified by X-ray crystallography.

In summary, the manuscript represents the description of an interesting anion and especially the theoretical work is well performed. However the experimental data do not allow the conclusions drawn.

Response to the comment: We thank this reviewer for the overall positive assessment of our work. Yet, we do not agree with the statement that the experimental data do not allow the conclusions drawn, as we do provide enough (direct or indirect) evidence for the identity of the compound, and – even more importantly – based on the combined experimental and theoretical work, we are 100% sure about the composition and charge of the anion, which is in the focus of interest of this whole study. So, we hope that the new changes served to dissipate all doubts about the composition of the compound, especially regarding the composition of the unique cluster anion.

Therefore I cannot recommend publication as long as the crystal structure determination has not improved. The authors should be encouraged to try other counter ions for crystallization or using single-crystal synchrotron data.

Response to the comment: We would like to note in advance that the crystallography has been done by a well-known expert in the field (Prof. Werner Massa), and that we have no doubts whatsoever that this is the best possible single-crystal structure analysis you can get from this compound. We tested many different crystals and measurement procedures (including different diffractometers), yet the intrinsic problem persists, as the statistical disorder of the cations leads to a "solid solution"-like behavior

(comparable to the situation in the solid state structure of fullerene, or in products of solvothermal or ionothermal syntheses). For this, we added further analytics to confirm the heavy atom composition of the compound except for the crypt ligands, all of which confirm the given formula.

Based on this intrinsic property of the compound, we did and do not agree with the reviewer's critical remarks on the single-crystal structure determination. On the contrary: given the nature of the compound, it is remarkable enough that we have got a structure of this quality (with reasonable R values)! Please note in addition that the back-Fourier-transform method ("Squeeze") routine was programmed for exactly such cases.

Moreover, as well-known in Zintl chemistry, one cannot easily exchange the counterions, as other counterions usually result in (a) no crystals or (b) crystals of other compounds (this was tested, of course!). We have added a corresponding statement to the **Supplementary Discussion**: "Attempts to improve the crystal quality by using other counterions did not help (as typical in Zintl chemistry); corresponding experiments with ammonium ions, phosphonium ions, alkaline earth cations, and the crown ether 18-crown-6 as a cation sequestering agent instead of crypt-222 failed in the formation of any crystalline material so far."

The intrinsic problem of orientational disorder in the structure will remain even when using synchrotron radiation. Though the absorption problems could be reduced, no substantial improvement is therefore expected, based on our long-standing experience.

We would like to thank the Editor and the Reviewers for giving us another chance to improve the narrative regarding the crystal structure analysis, and we hope that they acknowledge the changes made.

REVIEWERS' COMMENTS:

Reviewer #3 (Remarks to the Author):

The authors rely to a large extent on the crystallographic data. However, the revised version does not contain any new results on the experimental crystallographic data. In this context it is irritating that on one hand the authors conclude from "... 62.7% (including H atoms)" to the correct structure assignment, but at the same time state by this that 38.3 % are not assigned.

In addition it is irritating that a personal statement on one of the coauthors quality is pointed out: "We would like to note in advance that the crystallography has been done by a well-known expert in the field (Prof. Werner Massa), and that we have no doubts whatsoever that this is the best possible single-crystal structure analysis you can get from this compound."

In this context, it is important to read the crystallographers comment in the cif file: "Nevertheless, only the heavy atoms of the anion cluster could be localized and refined with anisotropic displacement parameters. Their unusually large but sensible values suppose orientational disorder over close positions in the structure. By the same reason, the cryptate ligands at K3 and K4 (on a mirror plane) could not be localized. As well, additional presence of en and/or toluene solvent molecules cannot be excluded." and especially the conclusion also given in the cif file: " Thus, the formula of the refined model does not correspond to the real composition of the crystal."

The data and arguments do not qualify the manuscript for publication in a high ranked journal. Therefore, I cannot recommend publication.

Reviewer #4 (Remarks to the Author):

This paper reports a Zn cluster which is novel due to its large cluster size (38 metal sites including 20 Zn) and low Zn valency. The structure contains disorder in both cationic and anionic parts, making the determination of compositions through crystallographic refinement less definitive. However, Such disorders are common in many interesting materials such as zeolite-type materials and MOFs. For metal-chalcogenide clusters, charge-balancing cations are also commonly disordered in molecular clusters (such as supertetrahedral T3-T4 clusters with size comparable to this K-Zn-Bi cluster) and 3-D frameworks. In most cases, the disordered parts are cavity-filling and charge-balancing, and the ambiguity in their determination (usually caused by orientational disorder) does not diminish the overall conclusion and the significance of the work, as is apparently the case here.

Reviewer 3 indeed raised some valid concerns, some of which are due to the authors' errors related to the formula in the cif file (the number of crypt-222) and the formula of the side product (the missing subscript 2 for [K(crypt-222)]). The authors have corrected such errors in the revised version and these

errors do not affect the original crystallographic refinement. Overall, the authors have adequately addressed the reviewer 3's comments.

Additional information on the stability of the compound in air and solvents and their solubility in different solvents might be useful for readers pondering the possible use of such chemistry (for example, the capture of such novel anionic clusters in a cationic MOF framework).

Also in Table S2, the calculated density is missing.

Reviewer's comments:

Reviewer #3:

The authors rely to a large extent on the crystallographic data. However, the revised version does not contain any new results on the experimental crystallographic data. In this context it is irritating that on one hand the authors conclude from "... 62.7% (including H atoms)" to the correct structure assignment, but at the same time state by this that 38.3 % are not assigned.

Response to the comment: We had responded to this issue in our previous letter.

In addition it is irritating that a personal statement on one of the coauthors quality is pointed out: "We would like to note in advance that the crystallography has been done by a well-known expert in the field (Prof. Werner Massa), and that we have no doubts whatsoever that this is the best possible single-crystal structure analysis you can get from this compound."

Response to the comment: In our view, it is important to state (in a responses letter only, of course) that experts were doing the tricky analytical work, not unexplored students, for instance. Werner Massa is an expert for crystallography (recently retired), and very well known in Germany as the author of a textbook, but may not be equally well known to any colleague across the globe. For this, we added this note for the reviewer's information.

In this context, it is important to read the crystallographers comment in the cif file: "Nevertheless, only the heavy atoms of the anion cluster could be localized and refined with anisotropic displacement parameters. Their unusually large but sensible values suppose orientational disorder over close positions in the structure. By the same reason, the cryptate ligands at K3 and K4 (on a mirror plane) could not be localized. As well, additional presence of en and/or toluene solvent molecules cannot be excluded." and especially the conclusion also given in the cif file: " Thus, the formula of the refined model does not correspond to the real composition of the crystal."

Response to the comment: First, we modified this comment to read: "Nevertheless, only the heavy atoms of the anion cluster (including K1 and K2) and the two crystallographically independent counterions (K3 and K4) could be localized and refined with anisotropic displacement parameters [...].", as this was actually not said clearly enough in the previous entry in the CIF. So, the reviewer might have missed the fact that we were able to localize all counterions and were therefore able to determine the overall charge of the cluster anion. Second, the sentence quoted in the last lines of the comment represents the facts: We do not list other than the refined atoms, yet know from our analytical results that the crystal must contain (heavily disordered) crypt-222 molecules. So, this is what needs to be said to explain the situation correctly.

The data and arguments do not qualify the manuscript for publication in a high ranked journal. Therefore, I cannot recommend publication.

Response to the comment: We respect this reviewer's reservations, yet at the same time, we are confident that the data we provide serve to confirm the identity of the compound, in particular the cluster anion, which we discuss in detail. This is in agreement with the assessment of all other reviewers, including Reviewer #4.

Reviewer #4:

This paper reports a Zn cluster which is novel due to its large cluster size (38 metal sites including 20 Zn) and low Zn valency. The structure contains

disorder in both cationic and anionic parts, making the determination of compositions through crystallographic refinement less definitive. However, Such disorders are common in many interesting materials such as zeolite-type materials and MOFs. For metal-chalcogenide clusters, charge-balancing cations are also commonly disordered in molecular clusters (such as supertetrahedral T3-T4 clusters with size comparable to this K-Zn-Bi cluster) and 3-D frameworks. In most cases, the disordered parts are cavity-filling and charge-balancing, and the ambiguity in their determination (usually caused by orientational disorder) does not diminish the overall conclusion and the significance of the work, as is apparently the case here.

Reviewer 3 indeed raised some valid concerns, some of which are due to the authors' errors related to the formula in the cif file (the number of crypt-222) and the formula of the side product (the missing subscript 2 for [K(crypt-222)]). The authors have corrected such errors in the revised version and these errors do not affect the original crystallographic refinement. Overall, the authors have adequately addressed the reviewer 3's comments.

Response to the comment: Thank you very much for this very positive assessment!

Additional information on the stability of the compound in air and solvents and their solubility in different solvents might be useful for readers pondering the possible use of such chemistry (for example, the capture of such novel anionic clusters in a cationic MOF framework).

Response to the comment: This is an important suggestion, which we were happy to follow.

Corresponding additions were placed in the “Methods” section on pages 11 and 12 of the revised manuscript.

On page 11: General synthesis details. All manipulations and reactions were performed under dry Ar atmosphere using standard Schlenk or glovebox techniques, as all Zintl compounds are sensitive to air and moisture.

On page 12: Compound 1 is soluble in dry en and DMF. Although the integrity of the highly-charged anion in solution could not be confirmed by means of ESI mass spectrometry (which may be a consequence of the corresponding measurement conditions), room-temperature solution ¹H NMR of **1** in DMF-d₇ indicate the presence of [K(crypt-222)]⁺ (see **Supplementary Figure 10**), hence corroborating the solubility as such.

Also in Table S2, the calculated density is missing.

Response to the comment: Thank you for pointing towards this accidental omission. The missing number was added to Supplementary Table 1 (page S13):

Z, ρ_{calc} [g cm ⁻³]	2, 2.365
---	----------

We would like to thank the Editor and the Reviewers once again for their efforts that were very helpful to improve the work!